# Agricultural management effects on mean and extreme temperature trends

Aine M. Gormley-Gallagher[1], Sebastian Sterl[1,2,3], Annette L. Hirsch[4], Sonia I. Seneviratne[5], Edouard L. Davin[5], Wim Thiery[1,5]

[1]Department of Hydrology and Hydraulic Engineering, Vrije Universiteit Brussel, Brussels, Belgium
[2]Department of Earth and Environmental Sciences, KU Leuven, Leuven, Belgium
[3]Center for Development Research, University of Bonn, Bonn, Germany
[4]ARC Centre of Excellence for Climate Extremes, University of New South Wales, Sydney, Australia
[5]Institute for Atmospheric and Climate Science, ETH Zurich, Zurich, Switzerland

*Correspondence to*: Aine M. Gormley-Gallagher (a.gormley@ulster.ac.uk)

**Abstract.**

Understanding and quantifying land management impacts on local climate is important for distinguishing between the effects of land management and large-scale climate forcings. This study for the first time explicitly considers the radiative forcing resulting from realistic land management and offers new insights on the local land surface response to land management. Regression-based trend analysis is applied to observations and present-day ensemble simulations with the Community Earth System Model (CESM) version 1.2.2 to assess the impact of irrigation and conservation agriculture (CA) on warming trends using an approach that is less sensitive to temperature extremes. At the regional scale, an irrigation- and CA-induced acceleration of the annual mean near-surface air temperature ($T_{2m}$) warming trends and the annual maximum daytime temperature (TXx) warming trends were evident. Estimation of the impact of irrigation and CA on the spatial average of the warming trends indicated that irrigation and CA have a pulse cooling effect on $T_{2m}$ and TXx, after which the warming trends increase at a greater rate than the control simulations. This differed at the local (subgrid) scale under irrigation where surface temperature cooling and the dampening of warming trends were both evident. As the local surface warming trends, in contrast to regional trends, do not account for atmospheric (water vapour) feedbacks, their dampening confirms the importance of atmospheric feedbacks (water vapour forcing) in explaining the enhanced regional

trends. At the land surface, the positive radiative forcing signal arising from enhanced atmospheric water
vapour is too weak to offset the local cooling from the irrigation-induced increase in the evaporative
fraction. Our results underline that agricultural management has complex and nonnegligible impacts on
the local climate and highlight the need to evaluate the representation of land management in global
climate models using climate models of higher resolution.

## 1 Introduction

According to observational and global climate model (GCM) data, temperatures associated with hot
extremes have increased consistent with global anthropogenic climate change (Sillmann and Croci-
Maspoli, 2009; Donat et al., 2013a, 2013b; Hartmann et al., 2013; Pendergrass and Hartmann, 2014;
Fischer and Knutti, 2015). However, hot spots of accelerated warming in annual maximum daytime
temperature (TXx) relative to local mean temperature ($T_{2m}$) simulated by climate models from phase 5 of
the World Climate Research Programme's (WCRP) Coupled Model Intercomparison Project (CMIP5)
are spatially inconsistent with observations (Donat et al., 2017). This is particularly the case over
southeast China, South America, north America and parts of Australia and Europe. In these regions, the
modelled TXx warming from the mid-twentieth century (1951–1980) to the late 20th/early 21st century
(1981–2010) was greater than the modelled $T_{2m}$ warming. In contrast to the models, the observations
showed that TXx warmed at a slower rate than $T_{2m}$. Further analysis of the CMIP5 ensemble over central
Europe by Vogel et al. (2018) highlighted that several GCMs overestimate the observed negative
correlation between summer precipitation and TXx, resulting in too strong future drying and associated
TXx increases under RCP8.5. This underlines the importance of a correct representation of land-
atmosphere coupling for simulating changes in temperature extremes at regional scales. These
discrepancies between multiple GCMs and observations raise the questions as to whether: (1) these model
results can be used to reliably project changes in local temperature extremes; (2) the discrepancies remain
if the rates at which warming occurs over a time period is examined, which less sensitive to outliers
common in extreme temperature data than the absolute temperature difference between two time periods,

as used in Donat et al. (2017); and (3) the inclusion of more processes that represent land-atmosphere coupling would enhance model skill.

Agricultural land management techniques, including irrigation and conservation agriculture, can have a cooling effect on hot temperature extremes (Davin et al., 2014; Hirsch et al., 2017; Thiery et al., 2017, 2020; Chen and Dirmeyer, 2019; Hauser et al., 2019; Jia et al., 2019). Irrigation diverts surface and groundwater resources to agricultural land to increase crop production (Fereres and Soriano, 2007). The addition of this water to the land surface is balanced by the loss of water via runoff, deep percolation, soil storage and/or evapotranspiration (ET) (Fereres and Connor, 2004). Under drier conditions, less evaporative cooling leads to amplified warming because the energy budget becomes dominated by sensible heating instead of latent heating (Donat et al., 2017). If irrigation water is added to the surface, this increases soil moisture as well as latent heat flux over the summer months, leading to more evaporative cooling at the land surface. This irrigation-induced surface cooling, in turn, challenges the radiative forcing concept, which assumes that as radiative forcing increases (from enhanced atmospheric water vapour) so too does surface temperature (IPCC, 2001; Boucher et al., 2004).

Conservation agriculture (CA), which involves crop residue management, crop rotation (Carrer et al., 2018; Lombardozzi et al., 2018) and minimal or no tillage (Kassam et al., 2015), can create climate feedbacks due to the presence of a crop residue over CA land that change both the radiative and hydrological properties at the surface (Davin et al., 2014). Hirsch et al. (2018) explored whether applying the no-till component of CA within the Community Earth System Model (CESM) improves the simulation of present-day climate. They found that the surface temperature response was influenced by three competing effects: (1) a surface albedo increase – which reduces the availability of energy for partitioning between the sensible and latent heat fluxes; (2) increased surface resistance (e.g. from mulch) – which reduces soil evaporation; and (3) increased soil moisture retention leading to enhanced transpiration. The local cooling response to CA was somewhat counteracted by grid-scale changes in climate over North America, Europe, and Asia because of negative atmospheric feedbacks. Grid-scale changes in climate counteracting local responses to land use change has also been demonstrated by Malyshev et al. (2015)

who showed that the subgrid signal of land use change in near surface temperature was diminished by the averaging with undisturbed portions of the pixels. The importance of local-scale responses to land cover change has also been indicated in observation-based studies (e.g., Mahmood et al., 2014; Li et al., 2015), yet few global-scale modelling studies examine the local land surface response to land management (Paulot et al., 2018; Meier et al., 2018).

Using GCMs, such as CESM, to simulate land-atmosphere interactions for investigating the effects of irrigation and agricultural conversion has been criticized as insufficient (Niyogi et al., 2002). This is partly because their coarse resolution (e.g., of order 100 km) hampers their performances in describing the present-day climate at the regional scale (Jiang et al., 2016). Furthermore, economic, societal and water resource factors are ignored – a void that initiated the so-called 'bottom-up' approach to evaluating the effects of land-use change (Douglas et al., 2006). Regarding the applicability of the knowledge produced by GCMs, they do not provide the skill required at the spatial scale to offer practical responses at the infrastructure scale (Hossain et al., 2015) or in terms of water resource management (Marshall et al., 2004). Despite these constraints, GCMs remain a prime tool for projecting changes in the climate system (Fajardo et al., 2020; Gupta et al., 2020; Hofer et al., 2020). Examples include the GCMs that are part of the latest Coupled Model Intercomparison Project (CMIP6) and used by the IPCC in consecutive assessment reports (Yazdandoost et al., 2021). However, these GCMs largely exclude agricultural management. In particular, no CMIP5 model incorporates irrigation or CA and only three CMIP6 models include irrigation, while none have CA. Pielke et al. (2011) suggested that landscape change is omitted from the CMIP5 models because the direct radiative impact of global landscape is a lower order than the radiative forcing from greenhouse gas emissions. This constitutes a reason to investigate their inclusion. That is, to distinguish between the effects of land management and other large-scale forcings such as rising $CO_2$ concentrations (Schultz et al., 2016), it is important to evaluate these processes in the GCMs and ultimately gain insight into the contrasts of impacts between regions under different climate regimes.

Considering the potential effects of irrigation and CA on climate (Thiery et al., 2017), it is possible that the discrepancies between climate models and observations regarding temperature changes (Donat et al., 2017) are because the models exclude the effect of agricultural management techniques on temperature. The goal of this study is thus to test the hypothesis that CESM version 1.2.2 overestimates warming trends in some regions because irrigation and CA are excluded. That is, warming rates are hypothesised to increase at a slower rate – showing signs of cooling, in irrigation- and CA-affected regions when climate models do account for a theoretical constant level of these land management practices. To realise this goal, the following objectives were formulated: (1) determine spatial warming rates using simulations that account for irrigation and CA and inspect whether CESM overestimates warming trends; (2) compare the observed rates of warming to the modelled rates of warming for irrigated and CA pixels, as well as non-irrigated and non-CA pixels; and (3) estimate the impact of irrigation on the spatial average of the warming rates over time for all land, selected regions, and irrigated and CA pixels. Within this framework, the novelty of this study lies in (i) an explicit focus on land management impacts on trends as opposed to the climatology; (ii) a comparison of the subgrid versus grid-scale response, offering important new insights on the local land surface land surface response to land management; and (iii) consideration of the radiative forcing resulting from realistic land management.

## 2 Materials and Methods

### 2.1 Irrigation and conservation agriculture implementation in CESM

To assess the influence of a theoretical constant level of either irrigation or CA on mean and extreme temperatures, we use the Community Earth System Model (CESM) version 1.2.2, which has contributed output to CMIP5 (Hurrell et al., 2013). The CESM atmospheric model was version 5.3 of the Community Atmosphere Model (CAM5.3) while the land surface model was version 4.0 of the Community Land Model (CLM4). Sea surface temperatures and sea ice fractions were prescribed from the data set described by Hurrell et al. (2008).

We analyse the control (1) and experimental (2) simulations presented in Thiery et al. (2017) for irrigation and in Hirsch et al. (2018) for CA. This set consists of three 5-member ensembles.

The first ensemble, the control (CTL), was set up to capture land-atmosphere components within a framework akin to that of the Atmospheric Model Intercomparison Project (AMIP). The period 1976-2010 was simulated with a horizontal pixel resolution of 0.9° latitude × 1.25° longitude. The first 5 years were discarded as spin-up, with trends evaluated for the period 1981-2010. On 1 January 1976, small random perturbations of $10^{-14}$ K were applied to the initial atmospheric temperature conditions. To focus on the influence of land–atmosphere interactions, rather than ocean–atmosphere feedbacks on the climate system, sea surface temperatures and sea ice fractions were prescribed from the data set described by Hurrell et al. (2008). Greenhouse gas concentrations were also prescribed from measurements, and satellite-based observations of vegetation phenology were imposed in CLM4.

The second ensemble, the irrigation (IRR) ensemble, follows an identical setup as the CTL experiment except that the interactive irrigation module in CLM4 was enabled. As described by Oleson et al. (2013), the irrigation parameterization in CLM4 divides the cropland area of each grid cell into non-irrigated and irrigated fractions corresponding to the portions that are equipped for irrigation – in accordance with Siebert et al.'s (2005) global map of irrigated areas (Figure 1a). The area of irrigated cropland in each grid cell is assigned as the smaller of the grid cells total cropland area and its area equipped for irrigation. What remains of the cropland area in the grid cell is regarded as non-irrigated cropland. It is important to note that implementation of transient irrigation was technically not possible in the CESM version 1.2, despite transient area equipped for irrigation data being available (Siebert et al., 2005), and therefore trends in the forcing are not considered.

The third ensemble, the CA ensemble, also follows the CTL experiment setup, but in this case the most likely distribution of CA was applied based on the CA dataset developed by Prestele et al. (2018). By splitting the existing CLM crop plant functional types (PFT) into a fraction under conservation agriculture and a fraction under conventional management, both forms of management are possible within a grid cell. Although the crop residue is assumed present all year, the implementation ensures that the increased soil

albedo effect on the total surface albedo is dampened during the growing season by the inclusion of canopy cover (Hirsch et al., 2018). Implementation of transient CA, however, was not possible due to data limitation as only a static CA map was available; hence we study a theoretical constant level of CA.

To examine heterogeneous influences within grid cells, subgrid tiles that represent local physical, biogeochemical, and ecological characteristics – and therefore local (subgrid) influences of irrigation and CA – were evaluated against regional (grid-scale) influences. Up to 21 surface tiles may occur within one grid cell in CLM4, including glacier, wetland, lake, urban, bare soil and 16 PFTs. For subgrid irrigation influences, all tiles are placed on one single soil column, except for the irrigated crop tile. Separating the soil columns in this way allows the soils underneath irrigated and rainfed crop tiles to have individual responses to atmospheric forcing (Schultz et al., 2016). Therefore, the subgrid-scale difference is the irrigated crop tile minus the rainfed crop tile. For subgrid CA influences, using the PFT-level outputs from CLM, it is possible to examine the subgrid-scale effect by subtracting the conventionally managed crop tiles from the CA crop tiles.

In addition, land masks were used to define and analyse: (1) all land pixels; (2) irrigated pixels only (where grid cells have a nonzero irrigated fraction); (3) CA pixels (the grid cells with a nonzero CA fraction) and (4) those regions of the Special Report on Managing the Risks of Extreme Events and Disasters to Advance Climate Change Adaptation (SREX) (IPCC, 2013) where irrigation and CA is extensive (Figure 1). The spatial points outside these masks as well as missing values in the observations were excluded (as 'NaN' values). These masks were applied to the investigations undertaken in this study. As the observational datasets (see below) were remapped to the model grid, this meant the same land masks (excluding Antarctica) could be used for each dataset.

### 2.2 Observational datasets

For evaluation purposes, observational datasets for annual mean $T_{2m}$ with a spatial resolution of $0.5° \times 0.5°$ for the same time period were obtained from the Climate Research Unit (CRU) (Harris et al., 2014). Annual mean TXx observational datasets were obtained from the daily Global Historical Climatology

Network extremes data set (GHCNDEX) (Donat et al., 2013a) and the Hadley Centre extremes data set (HadEX2) (Donat et al., 2013b) with a spatial resolution of $2.5° \times 2.5°$. These observational products were regridded to the CESM resolution using second-order conservative remapping (Jones, 1999). Thiery et al. (2017) and Hirsch et al. (2018) previously evaluated how the IRR and CA experiments alter the skill of CESM simulations (in terms of their agreement with observations). Thiery et al. (2017) demonstrated

that including irrigation has a small yet robust beneficial effect on the representation of TXx and $T_{2m}$ in CESM over irrigated and all land pixels. By including CA, Hirsch et al. (2018) showed a general improvement in the simulation skill over MED for TXx and $T_{2m}$ and enhanced skill for $T_{2m}$ over WNA, CNA, and CEU.

Observational data for the surface radiative temperature ($T_S$) at the subgrid scale were obtained from the E-OBS European CDG dataset for 1981-2010 over MED pixels. As a regional dataset, it has a higher spatial resolution and therefore enabled a skill of the models with respect to the local effects of land management. The E-OBS data were regridded to the CESM resolution using bilinear remapping.

**2.3 Statistical analysis**

The warming rate β was calculated using Sen's slope approach (Sen, 1968) based on the time and temperature values in each grid cell. This means that at each longitude and latitude point on land, there are 30 time measurements (1981-2010) with an associated temperature measurement (for each annual mean $T_{2m}$ and TXx). Therefore, there are 30x29/2 possible pairs of sample points, rendering 435 pairs for

each location.

Annual TXx and $T_{2m}$ values averaged across all land pixels and all irrigated pixels were computed for the CTL, IRR and CA ensemble means, as well as the GHCNDEX (TXx), HadEX2 (TXx) and CRU ($T_{2m}$) observations. A Sen's slope regression analysis was then carried out on the spatial mean temperatures of

TXx and $T_{2m}$ change over time (1981-2010) for (a) all pixels, (b) irrigated pixels and (c) CA pixels only, for both observations and the model ensembles.

The spatial mean warming rate across all (land or irrigated) pixels was also calculated. Additionally, all pixels within the SREX regions where irrigation is extensive (Thiery et al., 2017) –WNA, CNA, MED, WAS, SAS, SEA and EAS – were selected and their spatial means determined and examined. The SREX regions where CA is extensive (Hirsch et al., 2018) were also examined in greater detail. These include WNA, CNA, MED, SSA, CEU and SAU (Figure 1).

## 3 Results

### 3.1 Model Evaluation

First, we explore how the existing CESM climate simulation skill (i.e., how well the simulated and observed trends agree) is altered in IRR and CA relative to the skill obtained in the CTL. The model biases and spatial root mean square error (RMSE) values relative to the warming trends of the $T_{2m}$ and TXx global observational products are provided in Table 1. For the IRR ensemble, $T_{2m}$ warming trends are overestimated by $\sim$0.001 K yr$^{-1}$ across irrigated pixels, whereas over CA pixels $T_{2m}$ warming trends are overestimated by $\sim$0.002-0.004 K yr$^{-1}$ in both the CA and CTL ensemble. On average, the CTL, IRR and CA ensembles overestimate TXx warming trends by $\sim$0.007–0.03 K yr$^{-1}$ over all land pixels. Over irrigated pixels, the CTL and IRR ensemble overestimate TXx by $\sim$0.008–0.013 K yr$^{-1}$. Over CA pixels, the CTL and CA ensemble overestimate TXx by $\sim$0.006–0.013 K yr$^{-1}$. This means that while $T_{2m}$ warming rates have a slight low bias on average over all land and partially over irrigated areas, TXx warming trends are consistently too high over all land, irrigated and CA areas.

Second, to investigate how the uncertainty between the different irrigation and CA estimates of warming trends influences simulation skill, we examine the added value of including irrigation and CA for TXx and $T_{2m}$ over the regions where irrigation and/or the CA extent is greatest, as well as over global land, global irrigated land and global CA land (Figure 2). The added value is evaluated by calculating the absolute change (experiment minus control) in the spatial RMSE. Accounting for irrigation improves the simulation skill for trends over MED, WAS and SAS for $T_{2m}$ and over MED, WAS, SAS and SEA for

TXx (with HadEX2 as reference product). For WNA, CNA and EAS, the added value is negative or limited for both temperature metrics. Accounting for CA improves the simulation skill over CNA, CEU and SAU for the $T_{2m}$ and both TXx observational products and over the MED for the $T_{2m}$ and the TXx HaxEX2 observational products. For WNA, skill is reduced for all CA estimates. If we consider the grid cells where the land fraction within the CESM exceeds 50% ("all land") or just the grid cells that have a nonzero irrigation ("Irrigated land") is present, there is added value for $T_{2m}$ observational product over all land and the grid cells where irrigation has been applied. There is limited skill improvement for the TXx HadEx2 observational product. For the CA simulations, if we consider all land and the grid cells with a nonzero CA fraction ("CA land"), the model skill improves for the $T_{2m}$ observational product.

Third, we explore how the CESM climate simulation skill is altered in the subgrid-scale irrigation ($IRR_{SUB}$) and CA crop tiles ($CA_{SUB}$) relative to the skill obtained in the conventionally managed (CM) and rainfed crop tiles (RAIN) in the MED region. The model biases and spatial RMSE values relative to the warming trends of the $T_S$ observational product are provided in Table 2. For $IRR_{SUB}$, $T_S$ warming trends are overestimated by $\sim$0.004 K yr$^{-1}$ across irrigated MED pixels, which is an improvement in terms of bias when compared to the subgrid-scale data that does not account for irrigation (i.e., RAIN). However, according to the change in the spatial RMSE, accounting for irrigation does not improve the simulation skill for trends over MED irrigated pixels. This is likely because RMSE is more sensitive to outliers – whereas the bias is based on the spatial mean.

**3.2 Impact of Irrigation and Conservation Agriculture on Mean and Extreme Warming Trends**

Neither irrigation nor CA has a cooling effect on $T_{2m}$ and TXx warming rates in irrigated/CA or non-irrigated/CA regions (Figure 3 and Table 3). The results suggest a slight irrigation- and CA-induced acceleration of the annual $T_{2m}$ and TXx warming trends, rather than the hypothesised cooling. For instance, irrigation induced an increased $T_{2m}$ warming rate of 0.0023 K yr$^{-1}$ on average over land and 0.004 K yr$^{-1}$ across all irrigated pixels. To put these increases into context, the mean $T_{2m}$ CRU observed warming trend over irrigated pixels was 0.029 K yr$^{-1}$.


When the annual $T_{2m}$ and TXx temperatures are spatially averaged for each ensemble, the IRR and CTL simulations both overestimate the observed values for irrigated pixels (Figure 3a and 3b), and the CA and CTL simulations both overestimate the observed values over CA pixels (Figure 3c and 3d). However, the impact of irrigation and CA on the modelled spatially averaged temperatures improves the closeness to

that of the observations, i.e. there is an overall  there is an overall decrease in absolute temperature (Figure 3a-d), which aligns with current theory (Kueppers et al., 2007; Saeed et al., 2009; Kueppers and Snyder, 2012; Thiery et al., 2017, 2020; Hirsch et al., 2018).

What these results show in addition is, for the IRR and CA models – for all land, irrigated and CA pixels,

the spatially averaged $T_{2m}$ and TXx warming rates (the slopes) are higher than those of the CTL model. Therefore, rather than continuous cooling, there is evidence in Figure 4 of a pulse cooling phase during the spin-up years (Smith et al., 1998), after which the $T_{2m}$ and TXx warming trends increases at a greater rate than the control simulations.


In the case of CA, because crop residue is more likely to be applied during the summer/dry season (when TXx is typically recorded) to reduce evaporation (Figure 3l and Figure 5f), energy is shifted to the sensible heat flux (SHF) (Figure 3h and Figure 5j), increasing TXx (Figure 3d). The SHF response is not always consistent with the decrease in the latent heat flux (LHF) (Figure 5h), with some increases over Eastern

South America, Eastern North America, parts of Europe and Southeast Australia.

In the case of irrigation, the response also suggests two competing effects: (1) there is more water at the surface, so the energy budget shifts to the LHF (Figure 3i, Figure 5g and 5i), resulting in evaporative cooling (Figure 3a and 3b); and (2) because irrigation globally adds 418 $km^3$ $yr^{-1}$ of moisture to the

atmosphere (Thiery et al., 2017) and as water vapour acts as a greenhouse gas (GHG), it traps outgoing longwave radiation, radiating it back to the Earth's surface as downward longwave radiation (Figure TMQ), resulting in increased $T_{2m}$ and TXx warming trends (Figure 3a and 3b). The first effect appears more pronounced than the second due to the net cooling in Figure 3a and 3b. This means that despite the

water vapour (acting as a GHG) increasing downward radiation and the overall energy budget thus

increasing, most of it still goes to the latent heat flux leading to a net reduction in temperature (as compared to a situation without irrigation, where the sensible/latent ratio is more in favour of the latter). The limited warming effect of irrigation on atmospheric temperatures through water vapour forcing is consistent with earlier GCM studies inputting more than twice the amount of water vapour into the atmosphere through irrigation (32500 $m^3$ $s^{-1}$ or 1026 $km^3$ $yr^{-1}$) and finding limited radiative forcing

(Boucher et al., 2004; Sherwood et al., 2018).

We further investigate the potential warming of the Earth System irrigation-induced enhanced atmospheric water vapour by computing the top-of-atmosphere net radiation ($R_{n,TOA}$) in the CTL and IRR ensembles over the 1981-2010 period (Figure 4). As both ensembles employ prescribed, transient sea

surface temperatures, the difference in $R_{n,TOA}$ is a measure of irrigation-induced radiative forcing. The area-weighted global average $R_{n,TOA}$ is 0.4961 W $m^{-2}$ for the CTL ensemble (Figure 4a) and 0.5450 W $m^{-2}$ for the IRR ensemble. The radiative forcing from irrigation is therefore 0.0489 W $m^{-2}$, at least an order of magnitude smaller compared to other combined anthropogenic forcings over this period (IPCC, 2013) and consistent with previous estimates (Boucher et al., 2004; Sherwood et al., 2018). The positive

radiative forcing is mainly located over South Asia, and partially offset by negative forcing over central Asia, Greenland and Antarctica (Figure 4b). Breakdown of the irrigation-induced $R_{n,TOA}$ change into the shortwave and longwave components shows that the forcing is dominated by the longwave signal (+0.0583 W $m^{-2}$), with the shortwave signal even showing signs of a slight albedo increase (-0.0094 W $m^{-2}$), presumably from enhanced low-level cloud cover (Sherwood et al., 2018). The additional water

vapor in the atmosphere and associated longwave trapping in CESM can thus explain the small, positive radiative forcing contributing to Earth System warming and associated enhanced near-surface temperature trends in irrigated regions (Figure 3a-b), but at the land surface this signal is too weak to offset the local pulse cooling from the irrigation-induced increase in evaporative fraction.

## 3.4 Subgrid-Scale Impacts

Our results indicate a subgrid-scale cooling effect of irrigation on $T_S$ warming trends that is more distinct and spatially consistent over irrigated pixels than grid-scale effects (Figure 5a versus Figure 6a). $T_S$ warming trends on irrigated tiles are on average -0.008 K yr$^{-1}$ (-24%) lower than their rainfed counterparts, whereas the trends are on average 0.001 K yr$^{-1}$ (+11%) higher on the grid cell level over irrigated land (Table 3). The subgrid-scale influences of irrigation on ET rates over irrigated tiles were also pronounced as they increased by 0.653 mm yr$^{-1}$ in comparison to rainfed tiles (Figure 6c and Table 3). The subgrid-scale influences of CA on $T_S$ warming trends are smaller in comparison to irrigation, with only a 0.001 K yr$^{-1}$ (-3%) dampening of warming trends and ET rates increased by 0.083 mm yr$^{-1}$ (46%), relative to their conventionally managed counterparts (Figure 6b and 6d and Table 3).

The cooler warming trends from irrigation at the subgrid-scale (Figure 6a) occurs where the ET rate increases (Figure 6g) as well as the latent heat flux (Figure 6e), suggesting the cooling is due to an increase in the latent heat flux, which is consistent with Cook et al. (2015) and Thiery et al. (2017). The heightened grid-scale $T_S$ warming trends (Figure 5a) generally align with a greater TMQ flux (Figure 5c) and increased $T_{2m}$ warming trends over irrigated pixels (Figure 3a), which signifies the longwave radiation trapping potential of the additional atmospheric water vapour. As the impact on trends is small (e.g. $T_{2m}$ and $T_s$ warming trends increased, respectively, by 0.004 K yr$^{-1}$ and 0.001 K yr$^{-1}$ across irrigated pixels), the finding is in agreement with Sherwood et al. (2018) who showed that additional water vapour has a small impact on global warming potential mainly because it rains out before reaching the altitudes needed to significantly contribute to the greenhouse effect. These findings thus support the concept of radiative forcing and the proviso that, at the land surface, the water vapour signal does not offset local cooling from the irrigation-induced increase in evaporative fraction, as described for Figure 3 and 4 and previously proposed by Boucher et al. (2004). However, because the subgrid-scale $T_s$ trends, in contrast to grid-scale trends, are computed within the same ensemble and thus do not account for atmospheric (water vapour) feedbacks, the sign reversal of irrigation-induced impact on grid-scale and subgrid-scale $T_s$ trends

confirms the importance of atmospheric feedbacks (water vapour forcing) in explaining the increased grid-scale $T_s$ and $T_{2m}$ trends.

When spatially averaged, over all pixels, the $T_S$ warming trends at the subgrid-scale show no evidence of
a pulse cooling phase due to irrigation (Figure 7c), which is in contrast the results over irrigated pixels – where there is both a cooling effect on $T_S$ and a dampening of $T_S$ warming trends (Figure 7a). This contrast is likely due to a combination of the remote effects of irrigation, the larger contribution of natural variability and an increased relative contribution of other components when considering all land pixels (Puma and Cook, 2010; Cook et al., 2015; De Vrese et al., 2016; Thiery et al., 2017).

Regarding CA, the slight overall warming of $T_S$ temperatures (Figure 7b) as well as the increase in $T_S$ warming trends over CA pixels for the MED region (Figure 7f) is possibly because of the decrease in soil evaporation as a result of crop residue over CA land (Figure 5f), inhibiting energy partitioning from the SHF (Table 3). The cooling of $T_S$ temperatures over all land pixels (Figure 8d) and the slight decline in
$T_S$ warming trends over CA pixels (Figure 7b and Table 3), however, suggests that the effect of increasing surface albedo and thus reducing the solar energy absorbed by the surface is dominant. Additionally, the close correspondence between CA and CM (Figure 7b) may reflect that the temperature response spatially is both positive and negative depending on which mechanism dominates and therefore the spatial aggregation for all CA and all CM pixels globally loses this (Figure 7d).

## 4 Discussion

This study examined the hypothesis of whether excluding a theoretical constant level of irrigation and CA contributes to the overestimation of warming by an Earth System Model. A Sen's slope model was built and applied to ensemble simulations from the Community Earth System Model that include
irrigation parameterization to determine if there are spatiotemporal patterns and why they exist. This unexpectedly showed that warming trends are not dampened due to either irrigation or CA, except for the subgrid-scale effect of irrigation on the warming trends of $T_S$.

The key findings of this investigation are a net cooling effect of irrigation and CA on the modelled
spatially averaged $T_{2m}$ and TXx, but, rather than continuous cooling, the warming trends showed a pulse
cooling phase, after which the sensitivity to climatic change remains. Under irrigation, the opposing
effects are the result of: (1) evaporative cooling; and (2) atmospheric water vapour strengthening the
greenhouse effect. Under CA, the contrasting effects are due to: (1) cooling from a tillage-induced
increase in surface albedo; and (2) reduced soil evaporation due to the presence of crop residue, limiting
energy partitioning to the latent heat flux. At the subgrid-scale, there was both a cooling effect on $T_S$ and
in the dampening of warming trends. This implies that enhanced evaporative cooling is the dominant
driver of the subgrid-scale temperature trends.

Although this study was constructed with great care and built on a state-of-the-art modelling suite, several
future developments could improve understanding of the impact of irrigation and CA on climate. Firstly,
the quality of the model(s) could be improved by using transient irrigation and CA extents and new land
cover datasets from the 6[th] phase of the Coupled Model Intercomparison Project (CMIP6) (Lawrence et
al., 2016). In this study, a static irrigation map for the year 2000 was used for the whole simulation period.
This likely contributes to our results being conservative. If, for instance, irrigation expands over time, the
cooling effect may become stronger and thus affect the warming trends. Furthermore, the extent to which
the increase in surface albedo (i.e., the first competing effect of CA) affects the sensible and latent heat
fluxes partly depends on soil moisture, which too is not static. Also, CMIP6 experiments are based on
annual emissions, whereas CMIP5 was based on decadal emissions and CMIP6 models were updated
with irrigation-related features and land cover maps that incorporate irrigation and CA expansion over
time (Goddard et al., 2013; Miao et al., 2014; Boer et al., 2016; Meinshausen et al., 2017; Stouffer et al.,
2017). CMIP6 models may therefore improve the dynamics between irrigation, CA and climate change,
provided that they represent these land management techniques in their surface schemes.

The second consideration is that all simulations used in this study (5 control, 5 irrigation and 5 CA) were
from a single model. Ensembles completed as such with the same model but different simulations (i.e.

based on different initial conditions) characterise the uncertainty associated with internal climate variability only, while multi-model ensembles also account for the impact of model differences (Tebaldi and Knutti, 2007; Knutti et al., 2010). This limitation can impact cloud uncertainties. Hirsch et al. (2017) found that the CESM tends to produce large cloud feedbacks over Central Europe, Central North America,

North Asia, and South Asia when more energy is reflected at the surface. Irrigation-induced increases in latent heat fluxes led to more water vapor in the lower atmosphere, which generated low-level clouds (see also Sherwood et al., 2017). This limited shortwave radiation and hence the amount of energy available at the surface because the increased cloud cover reflected more downward shortwave radiation above the cloud layer, resulting in surface cooling. This was enhanced by a corresponding decrease in sensible heat

fluxes, reflecting the decrease in the amount of energy available at the surface and/or the increase in latent heating. The impact of cloud cover combined with land management change remains challenging to resolve. Therefore, this study should ideally be repeated with other models. Donat et al. (2017), for instance, conducted their study on 20 CMIP5 models, but these models did not incorporate irrigation and CA.


Thirdly, irrigation and CA are the only agricultural management practices considered in this study (and done so individually), whereas other agricultural management practices have been shown as impactful (Luyssaert et al., 2014; Erb et al., 2016, 2018). Trend analysis of integrated land management practices could affect the outcome if there is a lumped effect. Building an additional stochastic model could account

for variations in the distribution of the impact of land management practices on warming trends. This would enable sensitivity analyses to ascertain the relative importance of irrigation and CA to the total warming trends (based on all land management practices), as well as the relative contributions of the uncertainty sources (model input, parameter, structure) to the total uncertainty in the model output.

The final consideration is whether regression-based models are suitable for analysing changes in highly variable climate data, particularly annual extreme temperature data (von Storch, 2006). Essentially, the regression slope blends forced temperature change and variability, to provide an estimation of the temperature variation over time – within which variance can be lost due to noisy data. Whether the TXx

and $T_{2m}$ temperatures were first spatially averaged and then the slope retrieved or if each slope was estimated for each pixel and then the overall trends examined, the outcome remains. This is unsurprising considering that in the spatial averaging the noise contributions are averaged out, while the individual regression data suffers from the variance loss related to regression. However, when applied to over 60 years of observational data, the regression model used in this study showed similar trends to using the difference between the past and the present average temperatures (not shown). This implies that the irrigation and CA-inclusive climate system may require a longer timeframe (than the 30 years plus a 5-year spin-up period used) for trends to overtake the natural variability. Additionally, rather than aggregating all months, trends during individual months or seasons could be examined. This can affect, for instance, the influence of irrigation on $T_s$, which has a clear seasonal pattern, with more cooling during the driest and/or hottest months (Thiery et al., 2017). A smaller magnitude in TXx response to CA at the subgrid-scale has also been noted during the summer season due to a larger leaf area index (LAI) reducing soil surface exposure and thus the contrast between CA and conventionally managed crops (Hirsch et al., 2018). Furthermore, the implementation of CA within CESM does not capture crop planting and harvesting cycling (Davin et al., 2014), which would affect the LAI of the crop and potentially the effect of CA on surface climate.

## 5 Conclusion

In this study the impact of a theoretical constant level of irrigation and CA on warming trends in global climate and climate extremes was assessed for the period of 1981–2010 using the Community Earth System Model. A Sen's slope regression-based analysis was performed to compute spatial-explicit warming trends and spatially averaged warming trends. Insight into how modelled temperature is affected in its median by irrigation and CA over time was provided.

An irrigation- and CA-induced acceleration of the annual $T_{2m}$ and TXx warming trends was evident. Estimating the impact of irrigation and CA on the spatial average of the warming trends indicated that irrigation and CA have a pulse cooling effect on $T_{2m}$ and TXx, after which warming trends increased at a greater rate than the control simulations. This differed at the subgrid-scale under irrigation where surface

temperature cooling and the dampening of warming trends were both evident. Therefore, irrigation-induced evaporative cooling is a more dominant effect at the local level than the strengthening of the greenhouse effect at regional scales as a result of enhanced atmospheric water vapour.

A model evaluation demonstrated that the simulations accounting for irrigation and CA satisfactorily reproduce observed warming trends in $T_{2m}$, but not the trends in temperature extremes of TXx. This signifies that the GCMs have more trouble representing the greater variability in the extreme temperatures, compared to that of the mean annual temperature, and that the Sen's slope models are more suited to the blended variability inherent to annual mean temperatures.


The findings overall provide valuable context on how model complexity can impact the simulation of trends and emphasise the need for a more in-depth evaluation of the sensitivity of future climate projections to irrigation and CA-induced temperature changes. A sensitivity analysis, using transient irrigation and CA extents, as well as additional land management techniques, within coupled climate

models based on CMIP6 output, is recommended. In this way, the variance can be approximated and the relative contributions of the uncertainty sources to the total uncertainty in the model output, as well as the relative importance of irrigation and CA to the total warming trends, can be quantified and compared. If the fundamental uncertainties relating to model structure dominate, then a more detailed analysis than the regression approach used in this study is suggested. Furthermore, we encourage the community to

compare the coarser resolution results gained in this GCM study with higher spatial resolution models and for seasonal and monthly time periods. This will support decision-making on the incorporation of agricultural management processes in future GCM projects.

**Acknowledgments**

This study was supported by the LAMACLIMA project, part of AXIS, an ERA-NET initiated by JPI

Climate, and funded by BELSPO (BE, Grant No. B2/181/P1) with co-funding by the European Union (Grant No. 776608).

We thank Prof Piers Forster and Dr Chris Smith at the University of Leeds for their valuable discussions and insight on the theoretical outcomes of this project.

A. L. Hirsch is supported through funding from the Australian Research Council (ARC) Centre of
Excellence for Climate Extremes (CE170100023).

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

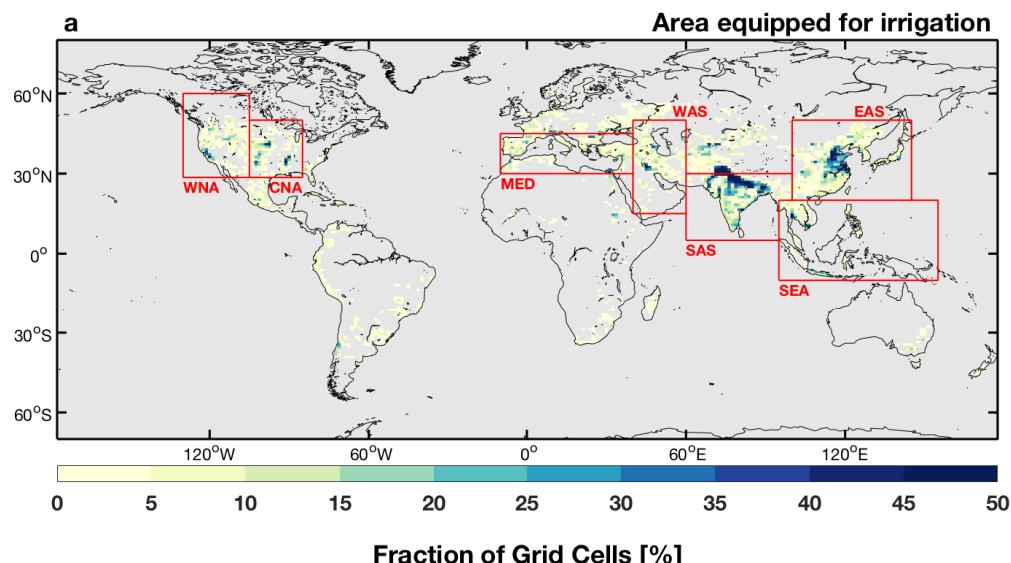

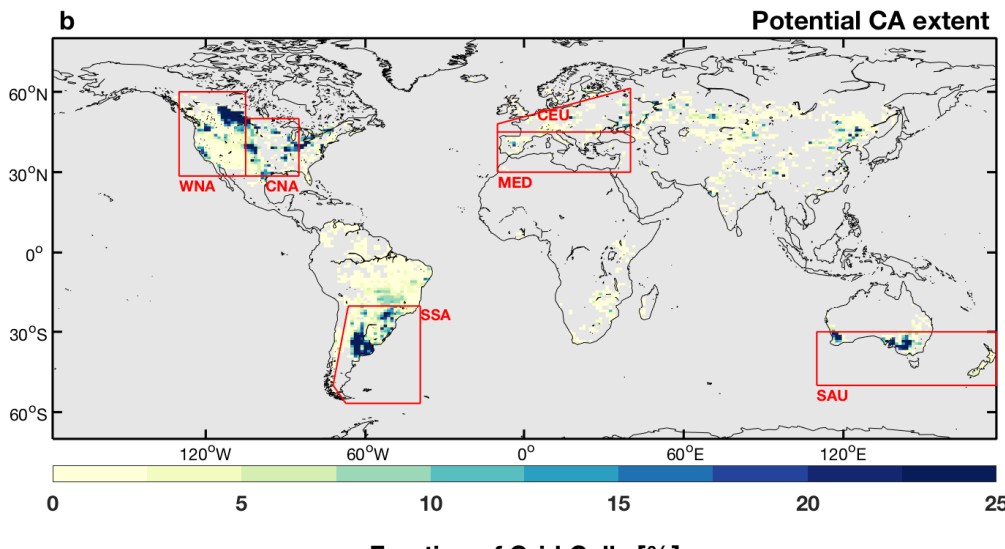

**Figure 1. (a) Percentage of each grid cell equipped for irrigation (%) (Siebert *et al.*, 2005). (b) Potential estimate of CA extent mapped to the CLM crop PFT (Prestele *et al*., 2018). The red boxes in (a) denote the regional domains where irrigation is extensive and were thus examined in greater detail including Western North America (WNA), Central North America (CNA), south Europe and Mediterranean (MED), West Asia (WAS), South Asia (SAS), Southeast Asia (SEA), and East Asia (EAS). The red boxes in (b) denote**
**the regional domains where CA is extensive and were thus examined in greater detail including WNA, CNA, MED, South-eastern South America (SSA), Central Europe (CEU) and Southern Australia (SAU).**

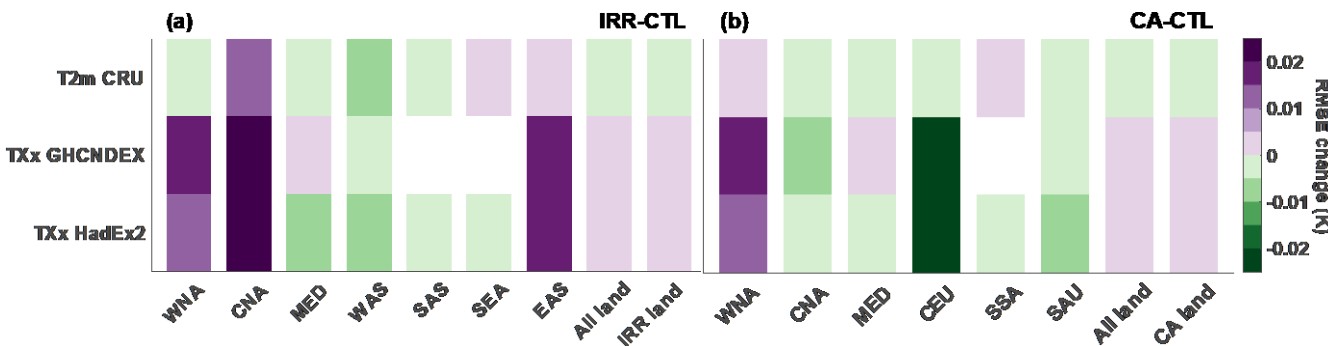


**Figure 2. Added value of including irrigation and CA in the simulated warming trends over 1981-2010. Absolute change in spatial root-mean-square error (RMSE) for the (a) IRR and (b) CA ensemble relative to the CTL ensemble over different regions (x axis) and with respect to 3 observational products (y axis). Considered regions are the SREX regions where irrigation is extensive (as highlighted in Figure 1a) and where CA is extensive (Figure 1b), in addition to global land, global irrigated land and global CA land.**
**Observational products are for near-surface air temperature $T_{2m}$ (CRU), annual maximum daytime temperature TXx (GHCNDEX and HadEX2). The spatial RMSEs are computed for the ensemble mean warming trend in every pixel, and subsequently averaged over the selected region. Regions with an observational coverage below 50% are marked in white.**


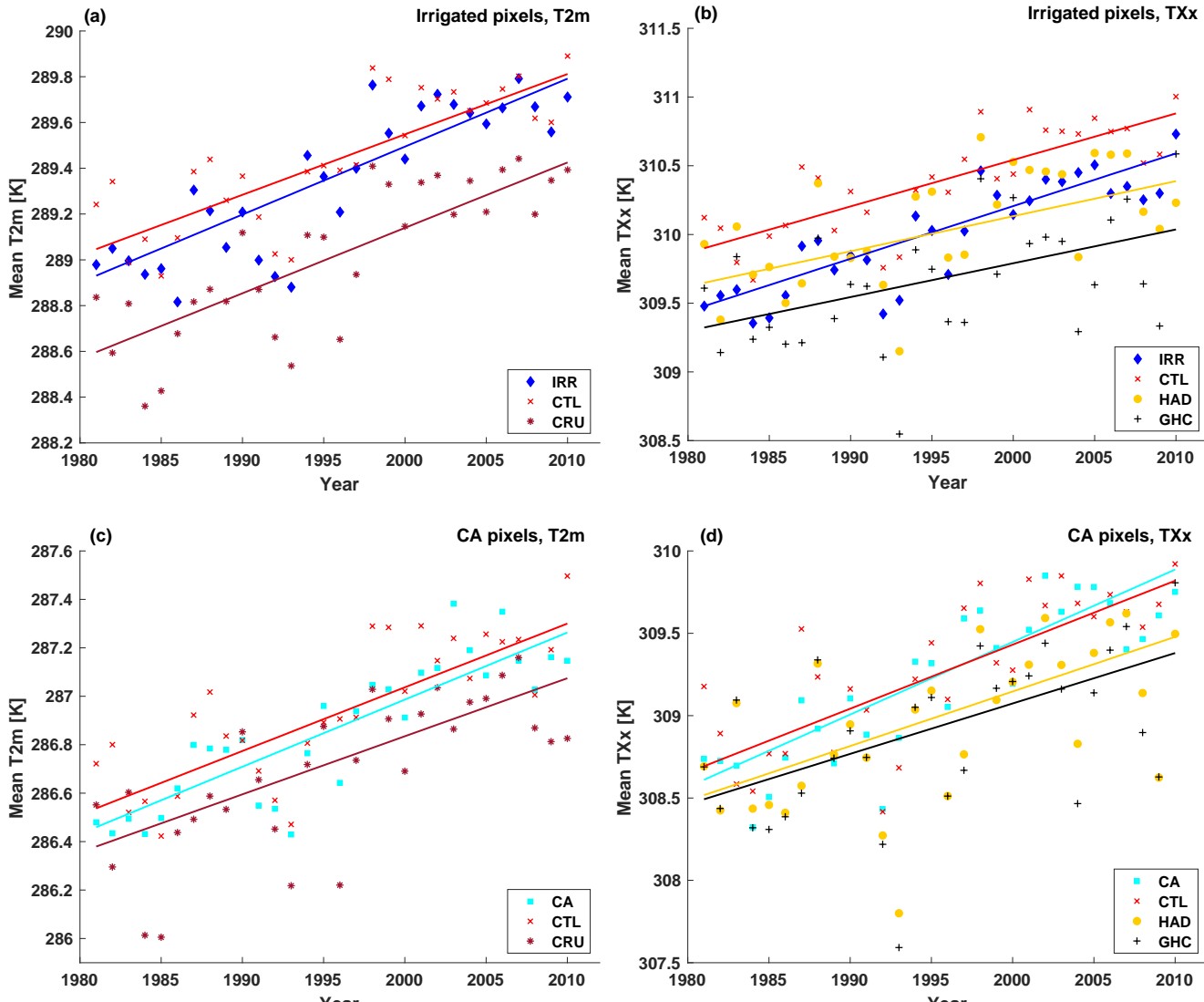

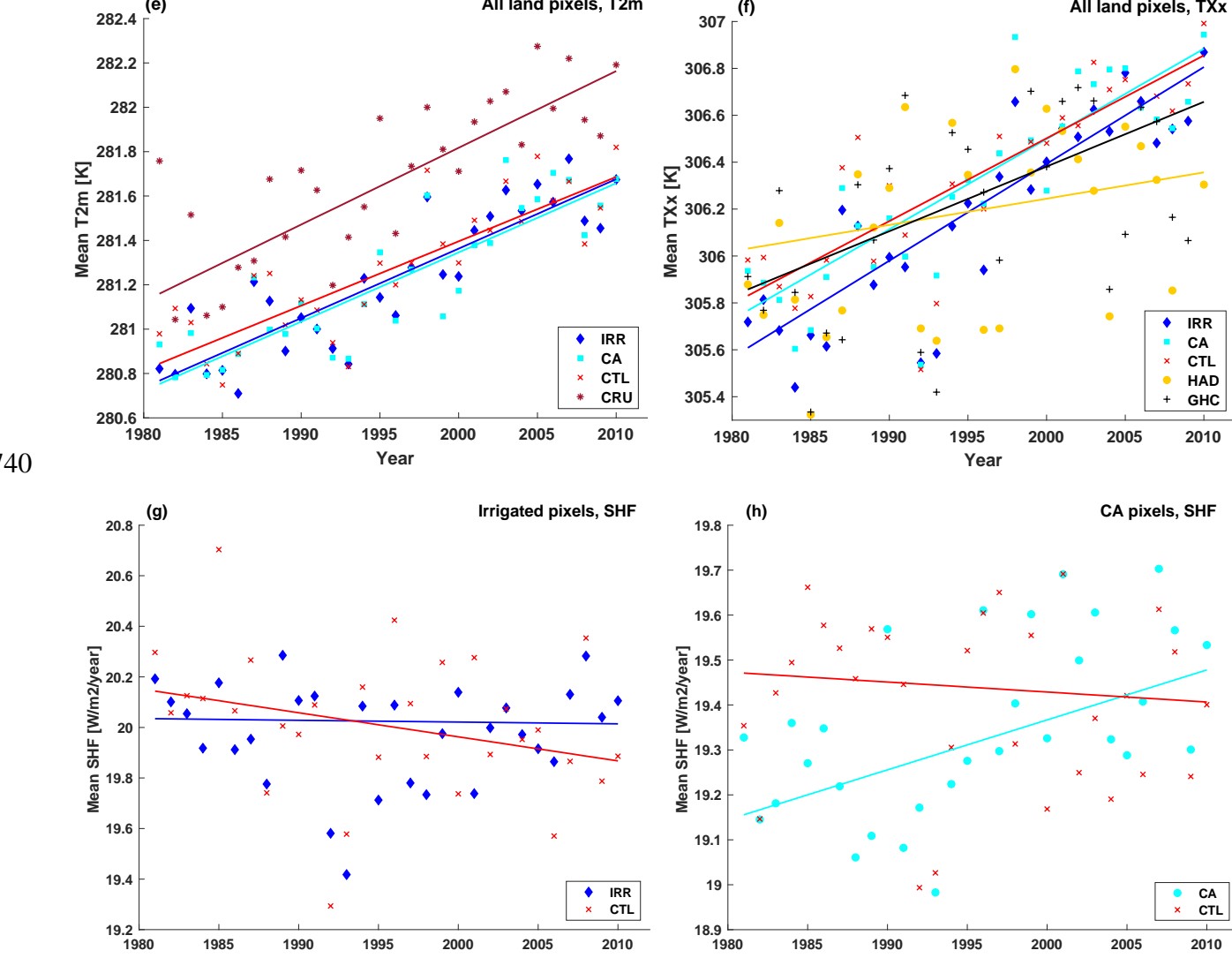


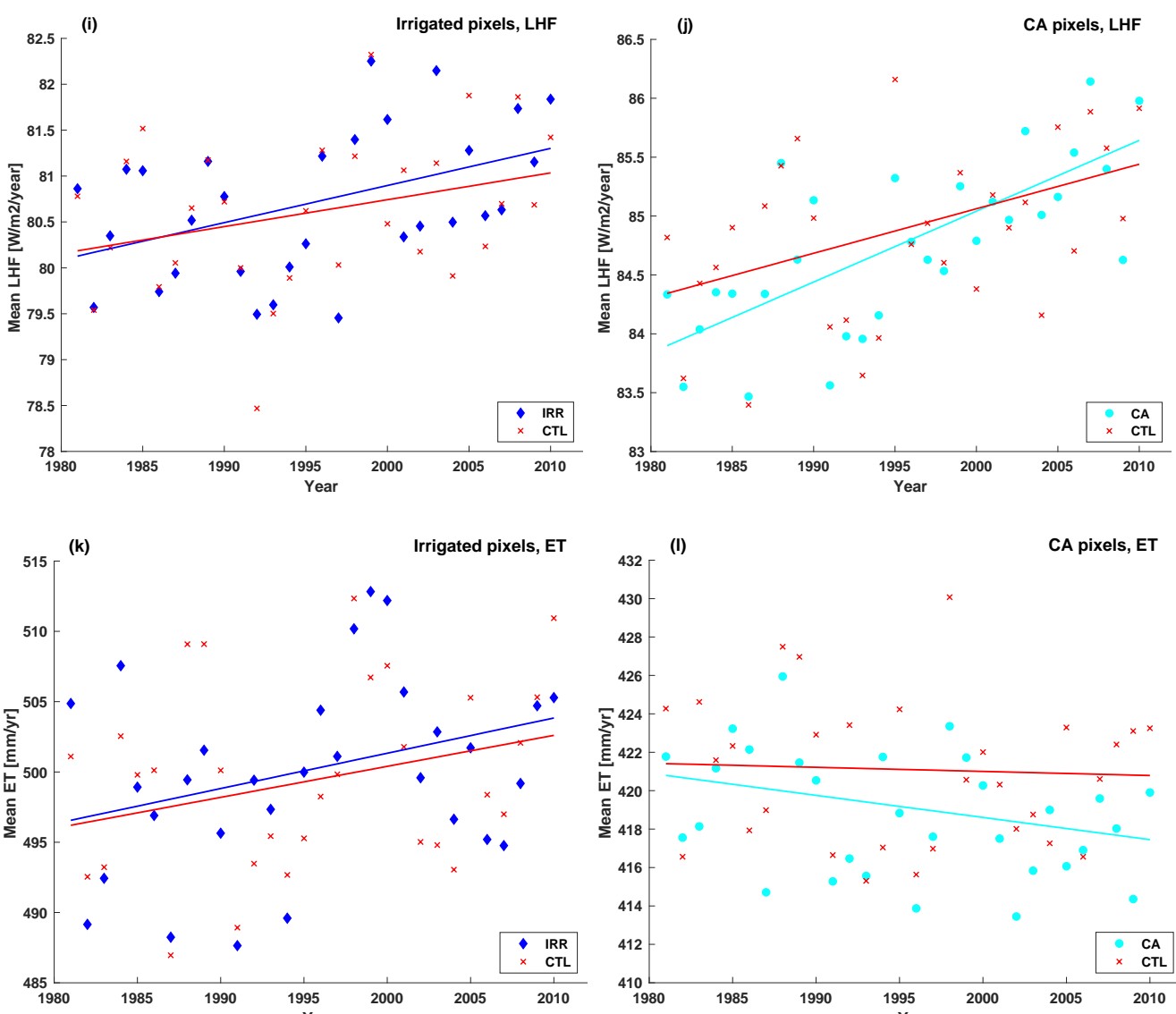


**Figure 3. Spatial average of the warming rates for T$_{2m}$ (a, c and e), TXx (b, d and f), SHF (g and h), LHF (i and j) and ET (k and l) for the CESM ensembles and observations. Data points specify the mean T$_{2m}$ and TXx temperatures, SHF and LHF and ET volumes for irrigated pixels (a, b, g, I and k), CA pixels (c, d, h, j and l), and (e-f) all land pixels. The slope was estimated using Sen's slope for the CTL (red), IRR (blue), CA (cyan), CRU (purple), HadEX2 (yellow), and GHCNDEX (black) temperatures.**


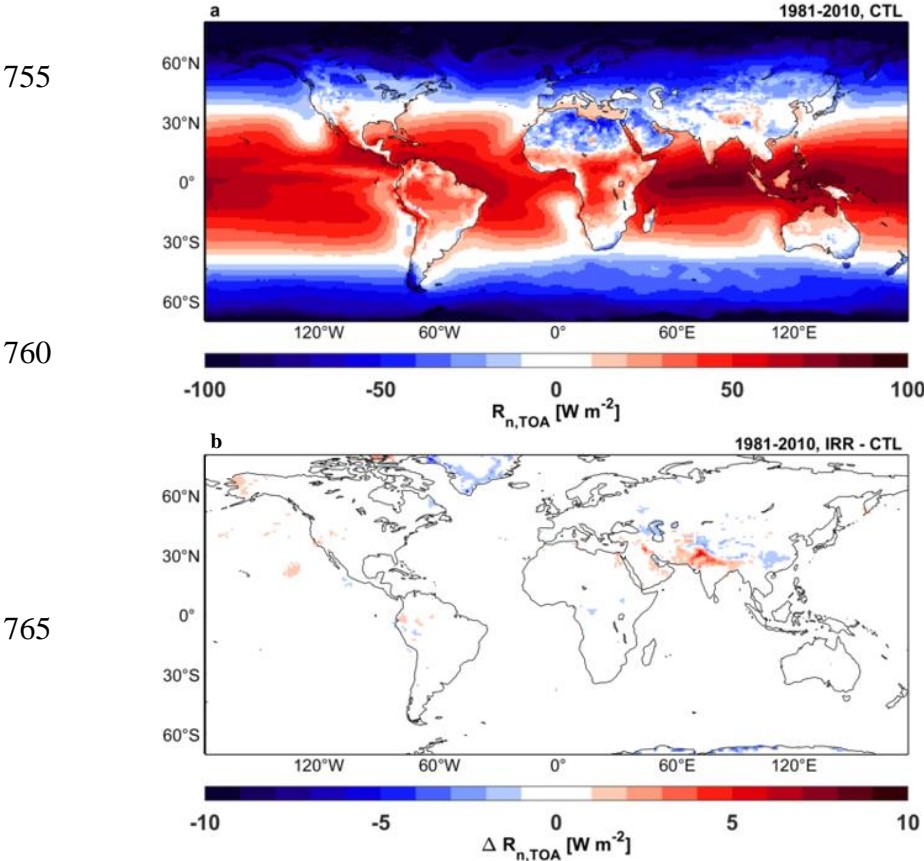




**Figure 4. Top-of-atmosphere (TOA) net radiation $R_{n,TOA}$ [W m$^{-2}$] in (a) the CTL ensemble. (b) Impact of irrigation on $R_{n,TOA}$. Difference map is based on the ensemble mean of each experiment for 1981–2010.**



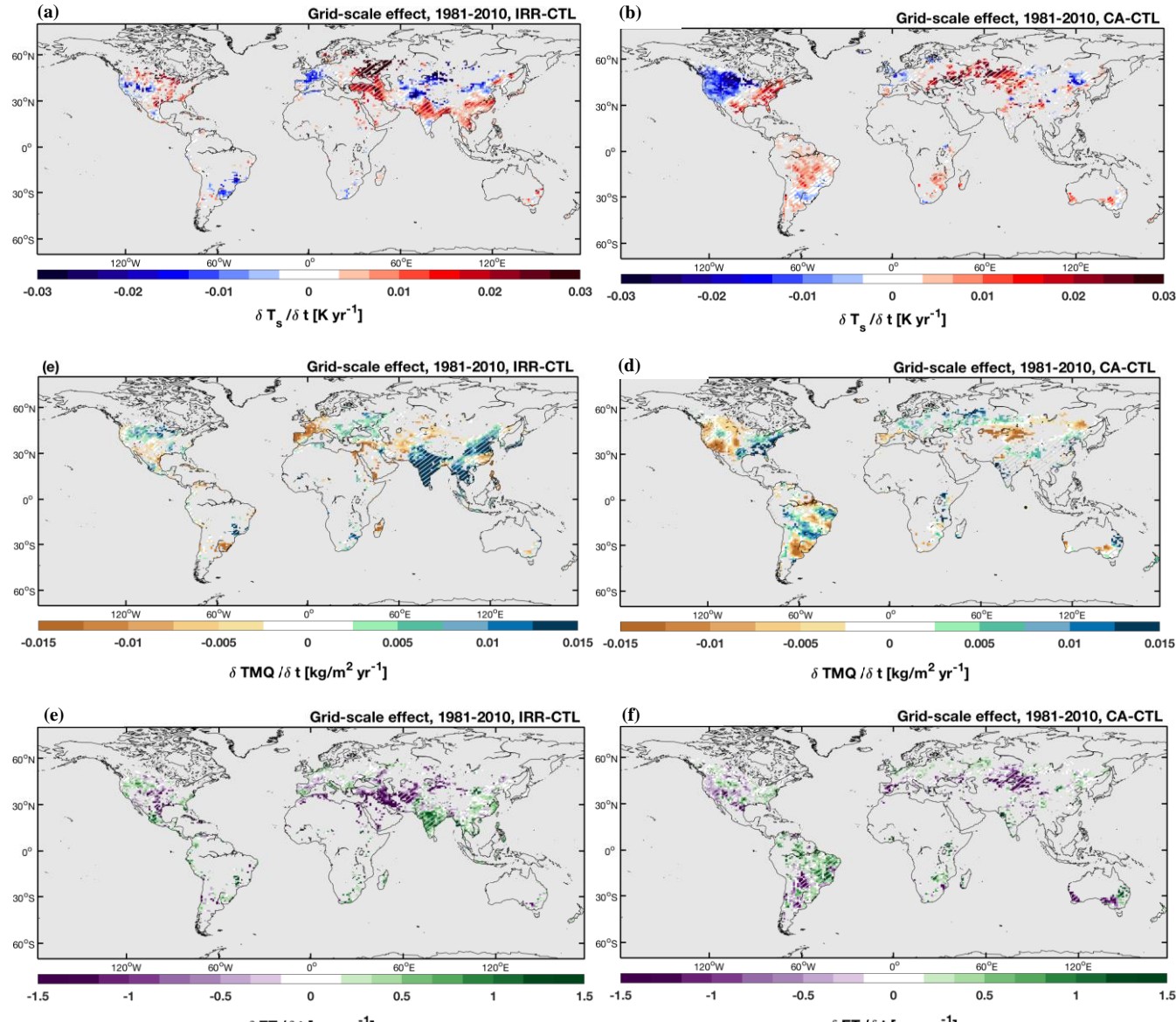

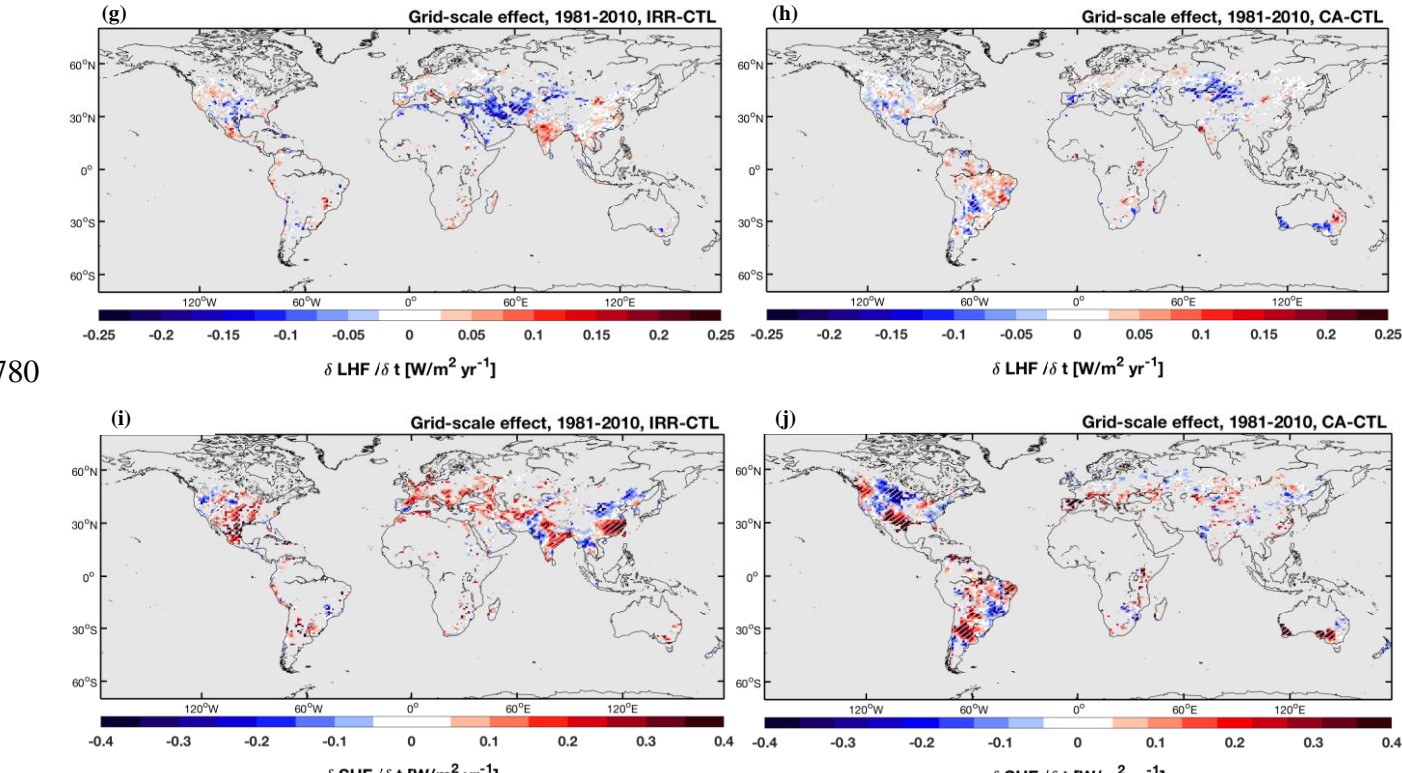

**Figure 5. Grid-scale differences between the CTL and IRR ensemble (IRR minus CTL) (a, c, e, g and i) and between the CTL and CA ensemble (CA minus CTL) (b, d, f, h and j). For $T_s$ (a-b), TMQ (c-d), ET (e-f), LHF (g-h) and SHF (i-j), displayed over irrigated/CA pixels for comparative purposes. Differences are based on the ensemble mean warming trends of each experiment for 1981–2010. Hatching denotes less than 10% change induced by the model on mean warming trends of lumped ensemble members.**



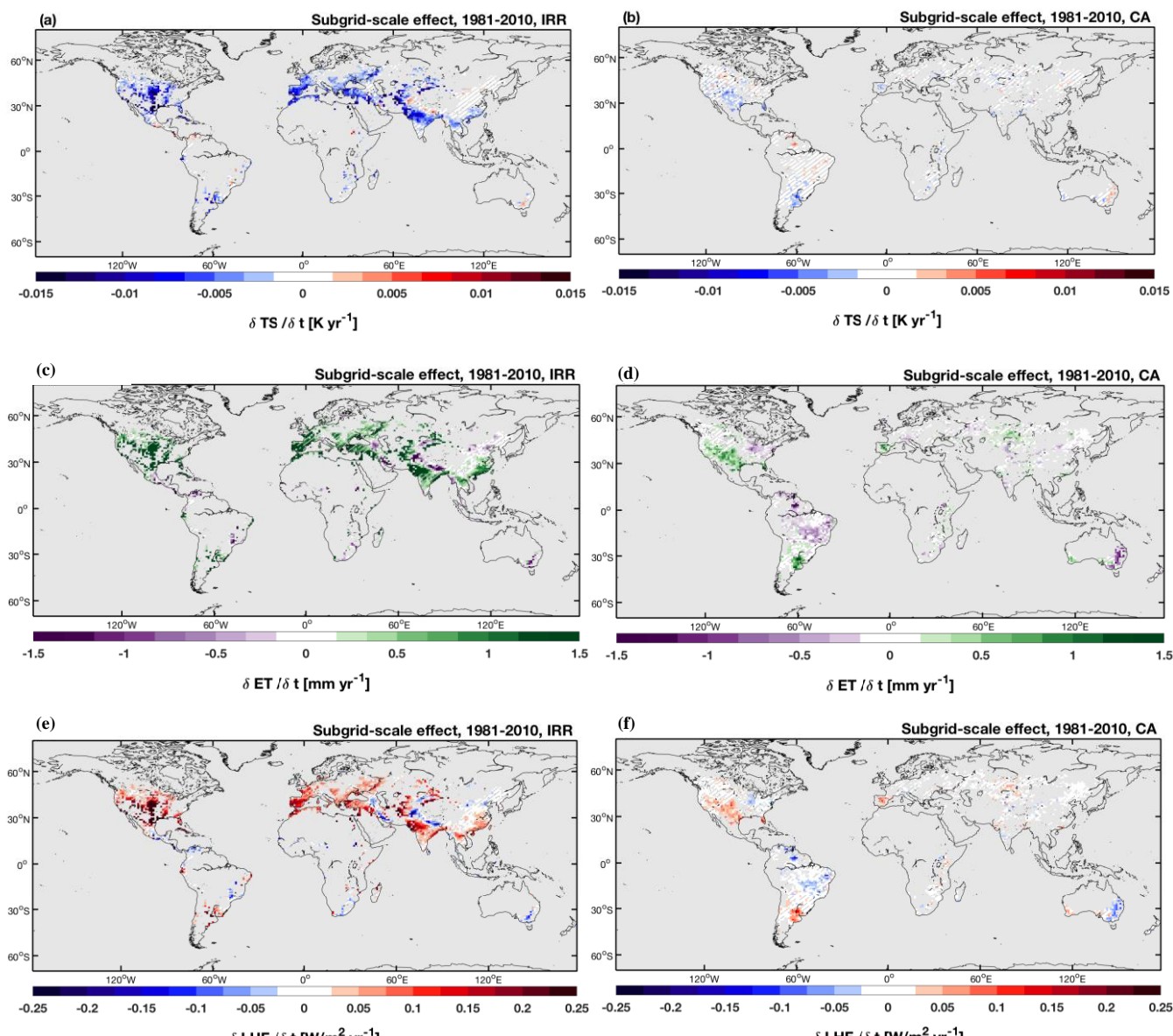

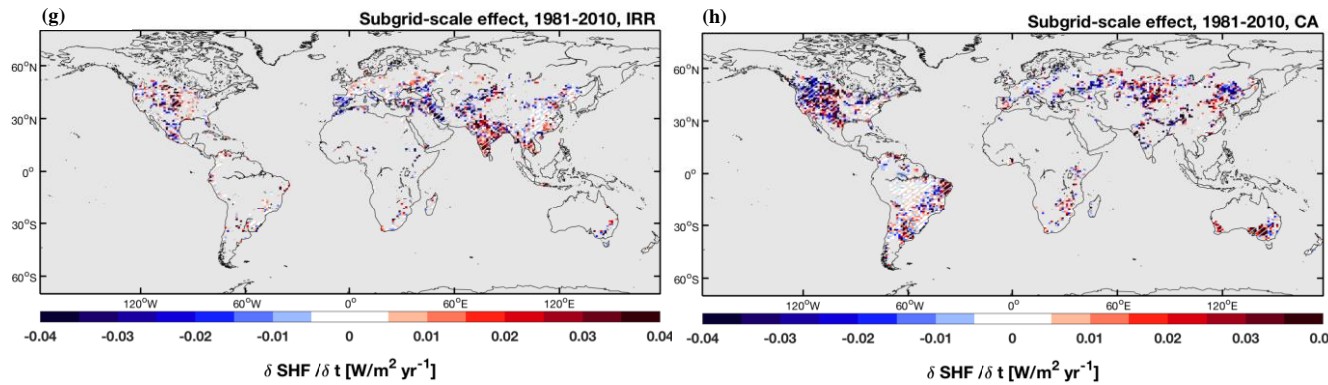


**Figure 6. Subgrid-scale differences between the irrigated and rainfed crop tile in the IRR ensemble (irrigated minus rainfed) (a, c, e and g) and between CA and conventionally managed (CM) crops (CA minus CM) (b, d, f and h). For T$_s$ (a-b), ET (c-d), LHF (e-f) and SHF (g-h), displayed over irrigated/CA pixels for comparative purposes. Differences are based on the ensemble mean warming trends of each experiment for 1981–2010. Hatching denotes less than 10% change induced by the model on mean warming trends of lumped ensemble members.**


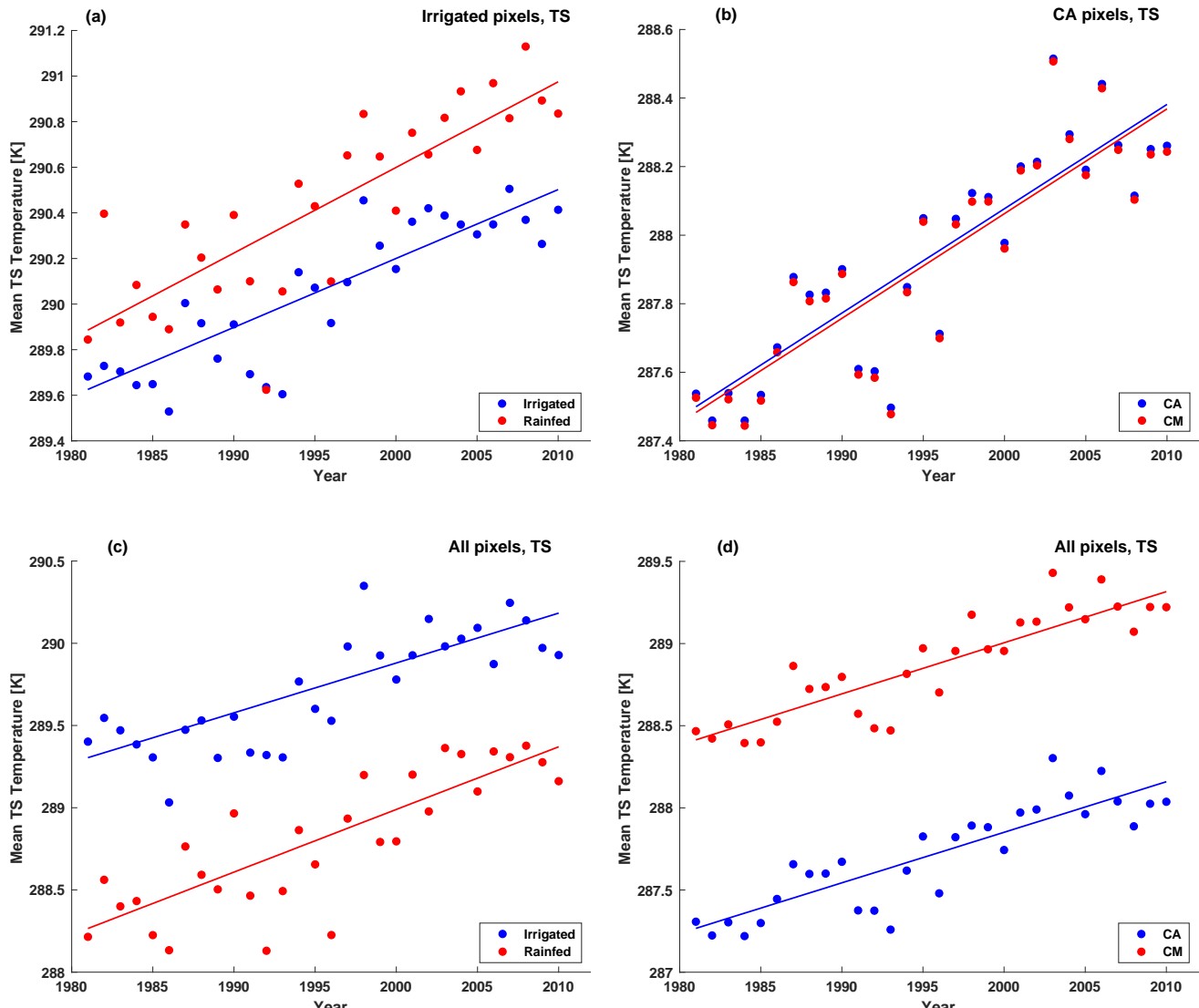

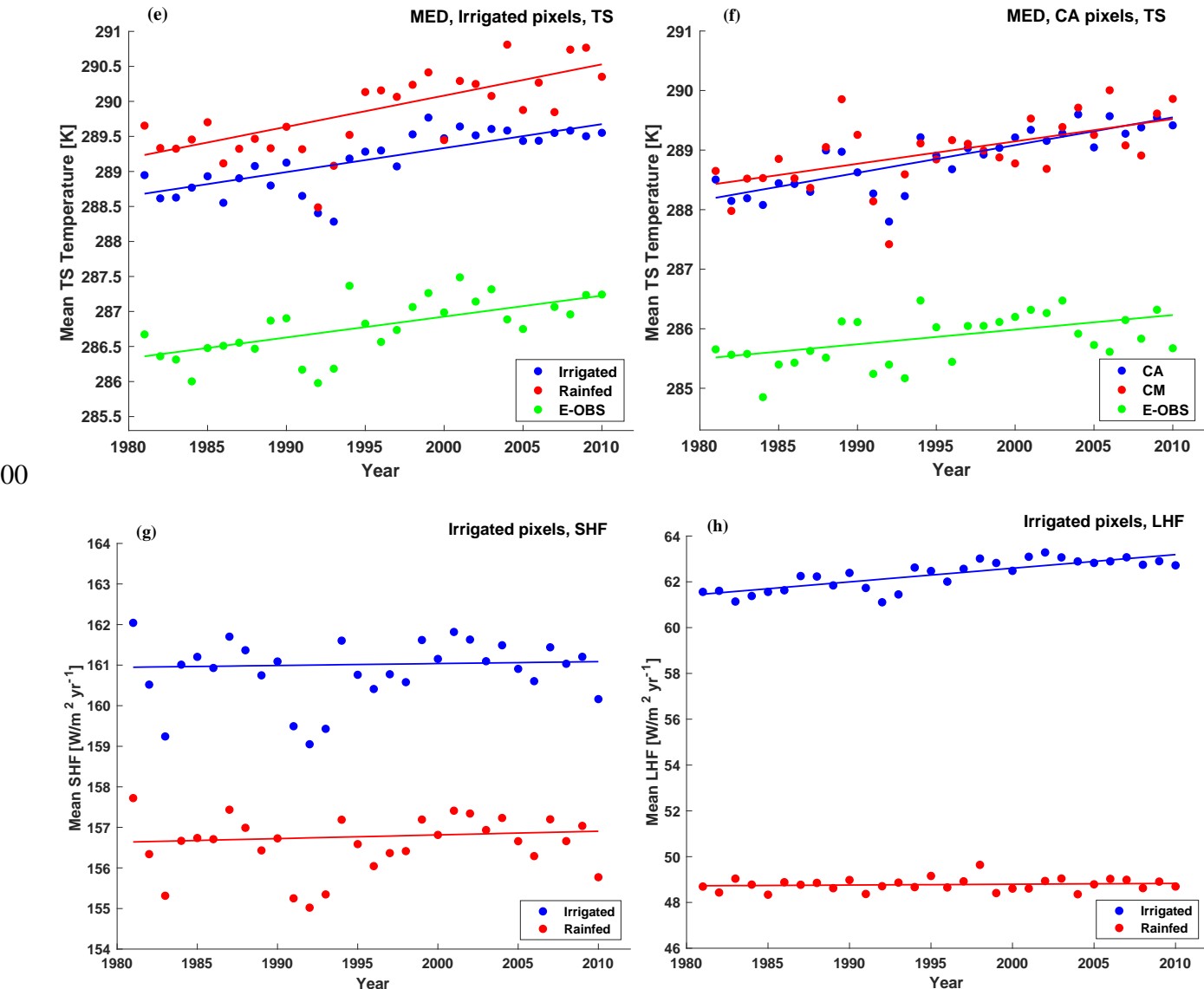


**Figure 7.** Average of the subgrid-scale warming rates for TS (a-f), the SHF (g) and LHF (h) over (a and g) irrigated pixels for the irrigated and rainfed crop tiles; (b and h) CA pixels for the CA and CM crop tiles; (b) all pixels for the irrigated and rainfed crop tiles; (d) all pixels for the CA and CM crop tiles; (e) irrigated pixels over the MED SREX region; and (f) CA pixels over the MED 805 SREX region. Data points are the mean TS, LHF and SHF values within the crop tiles and pixels specified. The slope was estimated using Sen's slope for the rainfed/CM (red), irrigated/CA (blue) experiments. For (a), (b), (c), (d), (e) and (f) the regions where less than 50% of the land pixels did not contain a value were excluded. For all land pixels (g and h), the minimum number of land pixels that needed to contain a value in order to be retained in the analysis was 15%.


**Table 1. Bias and spatial RMSE of the ensemble mean warming trends (slopes) of the CTL, IRR and CA experiments versus the observational products for the years 1981-2010[a].**

| Physical Quantity (Units) | All land bias | | | Irrigated land bias | | CA land bias | | All land RMSE | | | Irrigated land RMSE | | CA land RMSE | |
|---|---|---|---|---|---|---|---|---|---|---|---|---|---|---|
| | CTL | IRR | CA | CTL | IRR | CTL | CA | CTL | IRR | CA | CTL | IRR | CTL | CA |
| CRU $T_{2m}$ (K yr$^{-1}$) | -0.006 | -0.004 | -0.004 | -0.003 | 0.001 | 0.002 | 0.004 | 0.027 | 0.025 | 0.027 | 0.020 | 0.019 | 0.018 | 0.017 |
| GHCNDEX $TXx$ (K yr$^{-1}$) | 0.024 | 0.030 | 0.027 | 0.009 | 0.013 | 0.006 | 0.011 | 0.107 | 0.111 | 0.110 | 0.078 | 0.082 | 0.124 | 0.125 |
| HadEX2 $TXx$ (K yr$^{-1}$) | 0.007 | 0.013 | 0.010 | 0.008 | 0.012 | 0.008 | 0.013 | 0.135 | 0.135 | 0.136 | 0.121 | 0.122 | 0.086 | 0.087 |


[a]Regions with an observational coverage below 50% are excluded.

**Table 2. Bias and spatial RMSE of the subgrid-scale ensemble mean warming trends (slopes) of the RAIN, IRR$_{SUB}$, CA$_{SUB}$ and CM Experiments Versus the E-OBS (K yr$^{-1}$) observational product in the MED region for the years 1981-2010.**

| All MED pixels bias | | | | Irrigated MED pixels bias | | CA MED pixels bias | | All MED pixels RMSE | | | | Irrigated MED pixels RMSE | | CA MED pixels RMSE | |
|---|---|---|---|---|---|---|---|---|---|---|---|---|---|---|---|
| RAIN | CM | IRR$_{SUB}$ | CA$_{SUB}$ | RAIN | IRR$_{SUB}$ | CM | CA$_{SUB}$ | RAIN | CM | IRR$_{SUB}$ | CA$_{SUB}$ | RAIN | IRR$_{SUB}$ | CM | CA$_{SUB}$ |
| 0.032 | 0.033 | 0.022 | 0.035 | 0.015 | 0.004 | 0.013 | 0.022 | 0.040 | 0.039 | 0.031 | 0.026 | 0.028 | 0.031 | 0.027 | 0.026 |

**Table 3. Impact of irrigation and CA on various climatological values (absolute slope differences calculated as IRR minus CTL and CA minus CTL for grid-scale, IRR$_{SUB}$ minus RAIN and CA$_{SUB}$ minus CM for subgrid-scale) for the years 1981-2010[a].**

| | Physical Quantity (Units) | Irrigated Land | | | CA Land | | |
|---|---|---|---|---|---|---|---|
| | | CTL | IRR | ABS | CTL | CA | ABS |
| **Grid-scale** | $T_{2m}$ (K yr$^{-1}$) | 0.026 | 0.030 | 0.004[c] | 0.026 | 0.028 | 0.002[c] |
| | TXx (K yr$^{-1}$) | 0.034 | 0.038 | 0.004[c] | 0.039 | 0.044 | 0.005[c] |
| | $T_S$ (K yr$^{-1}$) | 0.009 | 0.010 | 0.001[c] | 0.016 | 0.015 | -0.001[c] |
| | LHF (W/m$^2$ yr$^{-1}$) | 0.029 | 0.041 | 0.012 | 0.029 | 0.053 | 0.024 |
| | SHF (W/m$^2$ yr$^{-1}$) | -0.010 | -0.001 | 0.009 | -0.010 | 0.004 | 0.014 |
| | Physical Quantity (Units) | RAIN | IRR$_{SUB}$ | ABS | CM | CA$_{SUB}$ | ABS |
| **Subgrid-scale[b]** | $T_S$ (K yr$^{-1}$) | 0.038 | 0.030 | -0.008[c] | 0.031 | 0.030 | -0.001[c] |
| | ET (mm yr$^{-1}$) | 0.286 | 0.939 | 0.653[c] | 0.182 | 0.265 | 0.083 |
| | LHF (W/m$^2$ yr$^{-1}$) | 0.004 | 0.060 | 0.056 | 0.009 | 0.014 | 0.005 |
| | SHF (W/m$^2$ yr$^{-1}$) | 0.009 | 0.005 | -0.004[c] | 0.056 | 0.060 | 0.004 |

[a]ABS denotes the absolute change of each given quantity.

[b]Regions with a coverage below 25% are excluded. For grid-scale calculations, regions with a coverage below 50% are excluded.

 [c]The changes significant at the 1% significance level (two-sided Wilcoxon signed rank test on ensemble mean slopes for irrigated/CA pixels.