# Peer review of "Agricultural management effects on mean and extreme temperature trends"

_Earth System Dynamics, 2020_

## Referee Comment (RC1) · Anonymous Referee #1 · 16 Aug 2020

This paper looks at the effects of prescribed representations of conservation agriculture and irrigation on mean annual 2m and maximum daytime temperature in CESM. There is some interesting analysis and potential for results that could be useful for the community. There are aspects of the sub-grid scale vs grid scale analysis, and possibility for critique of whether more processes enhance model skill, that are intriguing. The figures are generally well presented. However, there are several issues that need to be addressed.

The paper reads like a combination of previously published results (specifically, the ensembles used are already published in Theiry et al. (2017) and Hirsch et al. (2018)). That might be unfair, but the regression analyses is simple and it seems unlikely it wasn't done separately for CA and irrigation, and much of the explanatory analysis

references these two papers. It is the responsibility of the authors to show clearly why this is novel compared to what has come before.

The results (as shown in Table 1 especially) are difficult to reconcile with the statements made in the abstract and conclusions. Looking at Table 1, if the smallest RMSE (or the anomalies closest to zero) are considered, the Control simulation is better $\sim$ 2/3 of the time. The abstract says, "our results underline... the need to account for land management in climate projections". Surely the opposite is true, as the Control scenario does better by the measure most used to assess model skill. Even within the results, there appear to be contradictions. Line 218: "the impact of irrigation and CA on the modeled spatially averaged temperatures... is an overall cooling effect". Line 223: "for the IRR and CA models... the spatially averaged T2m and TXx warming rates are higher than those of the CTL model".

Some of the results are presented in such a way as to be somewhat misleading. For instance, the values in Figure 2 (% change in RMSE) with the colored categorization (which, being visual, is much stronger evidence to the reader than a table) can be compared with the equivalent anomaly in Table 1 (RMSE). For irrigation (IRR-CTL) the RMSE T2m (CRU) difference is -0.002, and the figure categorization is - 5-10%. For irrigation (IRR-CTL) the RMSE TXx GHCNDEX difference is +0.004, and the figure categorization is 0-5%. i.e. a difference in RMSE that is twice as big, is categorised as half the size in terms of color. This means that it looks as though the CA and IRR simulations are doing much better than if the simple RMSE is considered.

Some of the results are inconsistent with each other. For instance, on line 182, the range of temperature anomaly compared to observations is given as 0.007 - 0.03 in the text, but in Table 1 it is 0.007 – 0.024 (usually a number is rounded down when the last value is below 5). Or line 179 where the text says 0.004 for the Control, but Table 1 says 0.006. Figure 4b shows TXx HadEX2 on top of IRR and CNT much higher, but Table 1 shows CTL and IRR with differences from HadEX2 of 0.008 and 0.012 respectively.

The introduction does not do a good job of introducing the main point of the paper. The first paragraph sets up the issue that observations show less warming in TXx than T2m, but models get it the other way around (TXx warms more than T2m). But we basically don't hear about this issue again. Subsequent paragraphs in the introduction are brief summaries of key papers (by the authors) and do not provide the cohesive overview of each topic a reader needs in an introduction, instead being based around a particular reference.

The abstract doesn't do a good job of summarizing the paper. It goes straight to "we did things" without explaining why the reader should be interested, or what the relevance is, then goes to "it's important" without presenting the evidence of why it is important. It provides no context for what was done, then the emphasis on the "pulse cooling" in the abstract is not followed through in the results. The subgrid scale aspect seems to be the most novel part of the paper, but since the rationale for it isn't explained clearly in the methods or results, it's a minority of the results section, there's no comparison for the scale of the water vapor feedbacks, and no clear explanation of why the water vapor feedbacks are responsible for the differences between the grid scale and subgrid scale, it's not convincing.

The results section has paragraphs where the numbers in Tables are repeated, (with some unexplained deviations, as discussed above) with little attempt to give a view of what the results mean for the model's performance. There is no point in having a table if the text repeats what it says, and vice versa. There is absolutely a place for straightforward analysis and simple statements, but it needs to enhance clarity, not just be a list.

The Tables and Figures all need more detail in the captions, to help explain what they are and why they differ (and why those differences were deemed necessary). For instance: Table 1 – which years go towards the values? Table 2 – what are the "impact of irrigation and CA on various climatological values", because 0.026 K yr-1 for T2m doesn't make any sense as a number for the Control if it's supposed to be IRR – CTL as

described in the caption. Figure 4 – presumably when it says "average" it's the mean, but then line 378 says "median", which is a notably different average.

The results section has times where it would benefit from showing more evidence. For instance, the two paragraphs starting line 228 speculate that latent and sensible heat partitioning and changes in ET are responsible for the differences between CA and IRR, and the Control. But instead of exploring these and showing how the latent heat changes, there is just references to previous papers. I.e. it is not a result, it is a summary of previously published research. Similarly, the paragraphs line 289 – 315 contain a lot of speculation and references and not enough evidence.

The discussion section is missing, at the very least: cloud uncertainties (different models do them differently, and they are notoriously difficult to resolve, so that these results rely on them is problematic); uncertainties in the partitioning of latent and sensible heat when albedo changes (i.e. Bowen ratio); the fact that the CA increase in albedo is a huge assumption, as soil albedo is very heterogeneous and dependent partly on soil moisture (thus the CA modeled might be doing the wrong thing in many areas); the representation of transpiration in the model, and the fact that presumably the crop/vegetation cover is the same when in reality these changes would affect the LAI of the crop; the canopy interception and soil interception representation in the model, which affects the evaporation and thus how much the irrigation and CA affect the evaporation.

---

## Referee Comment (RC2) · Anonymous Referee #2 · 29 Dec 2020

Review of Agricultural management effects on mean and extreme temperature trends by Gormley-Gallagher et al for Earth System Dynamics Dec 28 2020

The authors' summary statement in the abstract is certainly an informative conclusion. They write

*"Our results underline that agricultural management has complex and nonnegligible impacts on the local climate and highlights the need to account for land management in climate projections."*

And further that

*"It remains challenging to resolve this, however, because it is difficult to separate land management from other effects in GCMs – particularly natural climate variability (Cook et al., 2015)".*

They summarize their paper with the text

*"The goal of this study is thus to test the hypothesis that CESM version 1.2.2 overestimates warming trends in some regions because irrigation and CA are excluded. That is, warming rates are hypothesised to decline – showing signs of cooling, in irrigation- and CA-affected regions when climate models do account for a theoretical constant level of these land management practices. To realise this goal, the following three objectives were formulated: (1) Determine spatial warming rates using GCM simulations that account for irrigation and CA and inspect whether CESM overestimates warming trends; (2) Compare the observed rates of warming to the modelled rates of warming for irrigated and CA pixels, as well as nonirrigated and non-CA pixels; and (3) Estimate the impact of irrigation on the spatial average of the warming rates over time (1981-2010) for all land, selected regions, and irrigated and CA pixels."*

However, the basis to quantify these impacts is flawed, or at least significantly muddled. First, model comparison studies are just model sensitivity studies. Without an assessment of model skill with the appropriate real world observed data, this is an incomplete (and potentially misleading) approach. The real world data needs to be on the spatial and temporal scale of the effect they are assessing (irrigation and conservation agriculture). The recent GRAINEX project quantified these scales [https://www.eol.ucar.edu/field_projects/grainex]. The model results should be compared against such data.

Indeed there are numerous regional, mesoscale and local studies that have assessed the role of irrigation and land management on weather and climate. The authors do not seem to be familiar with this research. Here are just a few

Adegoke, J.O., R.A. Pielke Sr., J. Eastman, R. Mahmood, and K.G. Hubbard, 2003: Impact of irrigation on midsummer surface fluxes and temperature under dry synoptic conditions: A regional atmospheric model study of the U.S. High Plains. Mon. Wea. Rev., 131, 556-564.

Betts RA. Implications of land ecosystem-atmosphere interactions for strategies for climate change adaptation and mitigation. Tellus B 2007, 59:602–615. doi:10.1111/j.1600-0889.2007.00284.x.

Boyaj et al, 2020: Increasing heavy rainfall events in south India due to changing land use and land cover. QJRMS https://doi.org/10.1002/qj.3826

Chen, C. J., C. C. Chen, M. H. Lo, J. Y. Juang, and C. M. Chang, 2020: Central Taiwan's hydroclimate in response to land use/cover change. *Env. Res. Lett.,* **15,** 034015

Douglas, E.M., D. Niyogi, S. Frolking, J.B. Yeluripati, R. A. Pielke Sr., N. Niyogi, C.J. Vörösmarty, and U.C. Mohanty, 2006: Changes in moisture and energy fluxes due to agricultural land use and irrigation in the Indian Monsoon Belt. Geophys. Res. Letts, 33, doi:10.1029/2006GL026550

He, Y., E. Lee, and J. S. Mankin, 2019: Seasonal tropospheric cooling in Northeast China associated with cropland expansion. Env. Res. Lett. 15, 034032.

Hossain, F., J. Arnold, E. Beighley, C. Brown, S. Burian, J. Chen, S. Madadgar, A. Mitra, D. Niyogi, R.A. Pielke Sr., V. Tidwell, and D. Wegner, 2015: Local-to-regional landscape drivers of extreme weather and climate: Implications for water infrastructure resilience. J. Hydrol. Eng., **10.1061/(ASCE)HE.1943-5584.0001210** , 02515002.

Marland, G., R.A. Pielke, Sr., M. Apps, R. Avissar, R.A. Betts, K.J. Davis, P.C. Frumhoff, S.T. Jackson, L. Joyce, P. Kauppi, J. Katzenberger, K.G. MacDicken, R. Neilson, J.O. Niles, D. dutta S. Niyogi, R.J. Norby, N. Pena, N. Sampson, and Y. Xue, 2003: The climatic impacts of land surface change and carbon management, and the implications for climate-change mitigation policy. Climate Policy, 3, 149-1

Pielke Sr., R.A., 2001: Influence of the spatial distribution of vegetation and soils on the prediction of cumulus convective rainfall. Rev. Geophys., 39, 151-177

Pielke Sr., R.A., R. Mahmood, and C. McAlpine, 2016: Land's complex role in climate change. Physics Today, 69(11), 40.

Ullah et al 2020: How Vegetation Spatially Alters the Response of Precipitation and Air Temperature? Evidence from Pakistan. Asian Journal of Atmospheric Environment 14(2):133 145

Woldemichael, A.T., F. Hossain, and R. A. Pielke Sr., 2014: Impacts of post-dam land-use/land-cover changes on modification of extreme precipitation in contrasting hydro-climate and terrain features. J. Hydrometeor., 15, 777–800, doi:10.1175/JHM-D-13-085.1

Zhang, T., R. Mahmood, X. Lin, and R.A. Pielke Sr.,2019: Irrigation impacts on minimum and maximum surface moist enthalpy in the Central Great Plains of the USA. Weather and Climate Extremes, 23, https://doi.org/10.1016/j.wace.2019.100197.

Unfortunately, the study does not have fine enough spatial resolution to realistically resolve these land use effects. As a result, the effects will likely be muted and quite possibly misrepresented. Even examining sub pixel (grid interval) model data is insufficient as local and mesoscale effects are missed.

As they report

*"The period 1976-2010 was simulated with a horizontal pixel resolution of 0.9° latitude × 1.25° longitude."*

This is much too coarse. Indeed since at least 4 grid increments are required to have some confidence that a feature is adequately resolved, their effective resolution is no finer than 3.6° latitude by 5° longitude.

Similarly, their observational analyses used to evaluate the model results are too coarse. They write

*"For evaluation purposes, observational datasets for annual mean T2m with a spatial resolution of 0.5° × 0.5° for the same time period were obtained from the Climate Research Unit (CRU) (Harris et al., 2014). Annual mean TXx observational datasets were obtained from the daily Global Historical Climatology Network extremes data set (GHCNDEX) (Donat et al., 2013a) and the Hadley Centre extremes data set (HadEX2) (Donat et al., 2013b) with a spatial resolution of 2.5° × 2.5° "*

And, as I mentioned above, even using sub-grid decomposition is significantly incomplete.  They write

*"To examine heterogeneous influences within grid cells, subgrid tiles that represent local physical, biogeochemical, and ecological characteristics – and therefore local (subgrid) influences of irrigation and CA – were evaluated against regional (grid-scale) influences. Up to 21 surface tiles may occur within one grid cell in CLM4, including glacier, wetland, lake, urban, bare soil and 16 PFTs."*

While useful in a model sensitivity study, its lack of connection to real world data for locations where actual irrigation and conservation agriculture are occurring is a serious oversight.

In their recommendations they write

*"The findings overall emphasise the need for a more in-depth evaluation of the sensitivity of future climate projections to irrigation and CA-induced temperature changes. A sensitivity analysis, using transient irrigation and CA extents, as well as additional land management techniques and climate models based on CMIP6 output, , is recommended."*

I agree with the first sentence. The second sentence, however, is incomplete as a necessary condition. Real world testing of the skill of the models with respect to how land management affects the weather and climate is required.  This must be completed using real world data that is on the appropriate space and time scales. This is not the case for this paper.

They also write

*"This will support decision-making when planning land management strategies that combine resource use efficiency with climate change adaptation and mitigation, enabling sustainable intensification of land management to meet mitigation targets and future demand for food, fuel, fibre, and water."*

The authors should be made aware that there are much more inclusive tools to assess sustainability. Sensitivity results from global models is, at best, a small part on the regional and local scales. Examples of such an approach are published in

Cross, M. S., et al. (2012). "The Adaptation for Conservation Targets (ACT) framework: a tool for incorporating climate change into natural resource management." Environmental Management 50(3): 341-351. DOI: 10.1007/s00267-012-9893-7

Hanamean, J.R. Jr., R.A. Pielke Sr., C.L. Castro, D.S. Ojima, B.C. Reed, and Z. Gao, 2003: Vegetation impacts on maximum and minimum temperatures in northeast Colorado. Meteorological Applications, 10, 203-215.

Hossain, F., E. Beighley, S. Burian, J. Chen, A. Mitra, D. Niyogi, R.A. Pielke Sr., and D. Wegner, 2017: Review approaches and recommendations for improving resilience of water management infrastructure:  The case for large dams. J. Infrastructure Systems, 23, Issue 4, Dec. 2017,  DOI: 10.1061/(ASCE)IS.1943-555X.0000370

Kittel, T.G.F., et al. (2011). "A vulnerability-based strategy for incorporating climate change in regional conservation planning: Framework and case study for the British Columbia Central Interior." BC Journal of Ecosystems and Management 12(1): 7-35.  http://jem.forrex.org/index.php/jem/article/view/89.

Kittel, T.G.F.  2013.  "The Vulnerability of Biodiversity to Rapid Climate Change."   Pp. 185-201 (Chapter 4.15), in: Vulnerability of Ecosystems to Climate, T.R. Seastedt and K. Suding (Eds.), Vol. 4 in: Climate Vulnerability: Understanding and Addressing Threats to Essential Resources, R.A. Pielke, Sr. (Editor-in-Chief).  Elsevier Inc., Academic Press, Oxford.  DOI:  10.1016/B978-0-12-384703-4.00437-8

Kling, M. M., Auer, S. L., Comer, P. J., Ackerly, D. D., & Hamilton, H. (2020). Multiple axes of ecological vulnerability to climate change. Global Change Biology, 26, 2798–2813

Ordonez, A., 2020. Points of view matter when assessing biodiversity vulnerability to environmental changes. *Global Change Biology*, *26*(5), pp.2734-2736.

Pielke Sr., R.A., R. Wilby, D. Niyogi, F. Hossain, K. Dairaku, J. Adegoke, G. Kallos, T. Seastedt, and K. Suding, 2012: Dealing with complexity and extreme events using a bottom-up, resource-based vulnerability perspective. Extreme Events and Natural Hazards: The Complexity Perspective Geophysical Monograph Series 196 © 2012. American Geophysical Union. All Rights Reserved. 10.1029/2011GM001086.

Romero-Lankao, P., et al. 2012: Vulnerability to temperature-related hazards: a meta-analysis and meta-knowledge approach. Glob. Environ. Change, http:// dx.doi.org/10.1016/j.gloenvcha.2012.04.002.

Stohlgren, T.J. and C.S. Jarnevich. 2009. Risk assessment of invasive species. In: M.N. Clout and P.A. Williams (eds.). Invasive Species Management: A Handbook of Principles and Techniques. New York: Oxford University Press. p. 19-35.

Thus, while I am pleased to see a study examining the effects of irrigation and conservation agriculture on climate, the study has significant shortcomings as summarized in this review.

---

## Author Comment (AC1) · 19 Feb 2021

We greatly thank the reviewer for the appreciation of the manuscript and for the constructive comments, which greatly helped to improve the quality of the study. Here below, we provide a point-by-point response to each comment. The modified manuscript text is shown in quotation marks.

1. The paper reads like a combination of previously published results (specifically, the ensembles used are already published in Theiry et al. (2017) and Hirsch et al. (2018)). That might be unfair, but the regression analyses is simple and it seems unlikely it wasn't done separately for CA and irrigation, and much of the explanatory analysis references these two papers. It is the responsibility of the authors to show clearly why

this is novel compared to what has come before.

Reply. We confirm that the analysis presented here is based on simulations that have been published previously. However, we believe that this new study moves beyond the state of the art in three ways. First, the main novelty of the current study lies in the explicit focus on trends, whereas previous studies focused on the influence of land management on the climatology (of means and extreme indicators). Second, the explicit focus on the subgrid versus grid-scale response offers important new insights on the local land surface land surface response to land management. Third, this study for the first time explicitly considers the radiative forcing resulting from realistic land management. We find that the positive radiative forcing signal arising from enhanced atmospheric water vapour is too weak to offset the local cooling from the irrigation-induced increase in the latent heat flux. This is been emphasised more clearly by including new evidence on the latent and sensible heat fluxes (warming/cooling trends as well as spatial averages) and the results and abstract adjusted to reflect these results as well as the reviewer's Point 2 and Point 6. The new abstract reads as:

Abstract. Understanding and quantifying land management impacts on local climate is important for distinguishing between the effects of land management and large-scale radiative forcings at the top of the atmosphere. This study for the first time explicitly considers the radiative forcing resulting from realistic land management and offers new insights on the local land surface response to land management. Regression-based trend analysis is applied to observations and present-day ensemble simulations with the Community Earth System Model (CESM) version 1.2.2 to assess the impact of irrigation and conservation agriculture (CA) on warming trends using an approach that is less sensitive to temperature extremes. At the regional scale, an irrigation- and CA-induced acceleration of the annual mean near-surface air temperature (T2m) warming trends and the annual maximum daytime temperature (TXx) warming trends were evident. Estimation of the impact of irrigation and CA on the spatial average of the warming trends indicated that irrigation and CA have a pulse cooling effect on T2m and TXx,

after which the warming trends increase at a greater rate than the control simulations. This differed at the local (subgrid) scale under irrigation where surface temperature cooling and the dampening of warming trends were both evident. As the local surface warming trends, in contrast to regional trends, do not account for atmospheric (water vapour) feedbacks, their dampening confirms the importance of atmospheric feedbacks (water vapour forcing) in explaining the enhanced regional trends. At the land surface, the positive radiative forcing signal arising from enhanced atmospheric water vapour is too weak to offset the local cooling from the irrigation-induced increase in the evaporative fraction. Our results underline that agricultural management has complex and nonnegligible impacts on the local climate and highlight the need to account to carefully represent and evaluate land management in climate models.

The new evidence, which has been added to the paper's Figure 6, show the Subgrid-scale differences between the irrigated and rainfed crop tile in the IRR ensemble and between CA and conventionally managed (CM) crops, for the latent heat flux (LHF) (k-l below) and the sensible heat flux (SHF) (o-p below). Grid-scale differences between the CTL and IRR ensemble and between CA and CM crops for LHF (m-n) and SHF (q-r) are also included over irrigated/CA pixels for comparative purposes, as detailed below. In addition, to the paper's Figure 7, data on the spatial average of the SHF (e) and LHF (f) warming rates for the irrigated and rainfed crop tiles over irrigated pixels, is now included, as shown in our Response Figure 2.

2. The results (as shown in Table 1 especially) are difficult to reconcile with the statements made in the abstract and conclusions. Looking at Table 1, if the smallest RMSE (or the anomalies closest to zero) are considered, the Control simulation is better âĹij 2/3 of the time. The abstract says, "our results underline. . . the need to account for land management in climate projections". Surely the opposite is true, as the Control scenario does better by the measure most used to assess model skill. Even within the results, there appear to be contradictions. Line 218: "the impact of irrigation and CA on the modeled spatially averaged temperatures. . . is an overall cooling effect". Line

223: "for the IRR and CA models. . . the spatially averaged T2m and TXx warming rates are higher than those of the CTL model".

Reply. The CTL simulation performs when considering the extreme (TXx) temperature results but not when considering mean temperatures (T2m). Particularly in the case of T2m for irrigation, the results in Table 1 show all cases are better all of the time. In one case for CA (all land), the RMSEs are equivalent, but otherwise the CTL RMSE is higher. So we disagree that the opposite is true in general, but we do agree that the statement in the abstract can be refined. It is also an opportunity for raising a critique/reflection point on whether more processes enhance model skill for the mean but not for extreme temperature, which adds to the addressing of the reviewer's introduction paragraph as well as the final point raised regarding the discussion. Therefore the abstract has been adjusted (see the new relevant statement below as well as the full abstract under our response to the reviewer's Point 1) as well as new results provided (also under reviewer's Point 1) so to better reconcile the results with the conclusion and abstract. The adjusted abstract statement reads as follows:

"Our results underline that agricultural management has complex and nonnegligible impacts on the local climate and highlight the need to carefully represent and evaluate land management in climate models."

Regarding the apparent contradiction between lines 218 and 223, the 'cooling effect' noted in line 218 (Figure 4) refers to a decrease in absolute temperature, that is, the intercept of the regression. Line 223, on the other hand, does refer to the trend over time, that is, the slope of the regression. This distinction has now been made clear in the text, which now reads:

"However, the impact of irrigation and CA on the modelled spatially averaged temperatures improves the closeness to that of the observations, i.e. there is an overall decrease in absolute temperature (Figure 4a-d), which is consistent with current theory (Kueppers et al., 2007; Saeed et al., 2009; Kueppers and Snyder, 2012; Thiery et al.,

2017, 2020; Hirsch et al., 2018)."

3. Some of the results are presented in such a way as to be somewhat misleading. For instance, the values in Figure 2 (% change in RMSE) with the colored categorization (which, being visual, is much stronger evidence to the reader than a table) can be compared with the equivalent anomaly in Table 1 (RMSE). For irrigation (IRR-CTL) the RMSE T2m (CRU) difference is -0.002, and the figure categorization is - 5-10%. For irrigation (IRR-CTL) the RMSE TXx GHCNDEX difference is +0.004, and the figure categorization is 0-5%. i.e. a difference in RMSE that is twice as big, is categorised as half the size in terms of color. This means that it looks as though the CA and IRR simulations are doing much better than if the simple RMSE is considered.

Reply. We agree with this point and are thus presenting the absolute change in the RMSE data (in K) in Figure 2 of the paper, as detailed in our Response Figure 3.

4. Some of the results are inconsistent with each other. For instance, on line 182, the range of temperature anomaly compared to observations is given as 0.007 - 0.03 in the text, but in Table 1 it is 0.007 – 0.024 (usually a number is rounded down when the last value is below 5). Or line 179 where the text says 0.004 for the Control, but Table 1 says 0.006. Figure 4b shows TXx HadEX2 on top of IRR and CNT much higher, but Table 1 shows CTL and IRR with differences from HadEX2 of 0.008 and 0.012 respectively.

Reply. The 0.007 - 0.03 in the text refers to the range of temperature anomaly for all three (CTL, IRR and CA) experiments, not just the CTL, which is consistent with the data in Table 1 (the 0.03 K yr-1 TXx bias is noted for the GHCNDEX observations and the IRR ensemble). This has now been clarified in the text, as follows:

"On average, the CTL, IRR and CA ensembles overestimate TXx warming trends by âĹij0.007–0.03 K yr-1 over all land pixels. Over irrigated pixels, the CTL and IRR ensemble overestimate TXx by âĹij0.008–0.013 K yr-1. Over CA pixels, the CTL and CA ensemble overestimate TXx by âĹij0.006–0.013 K yr-1."
For the reviewer's point regarding line 179, indeed the text referring to CTL T2m bias should read according to the data in Table 1 – i.e. 0.006. In order to also address the reviewer's point 7 below, we have addressed this by not including the statement in the text that specifies the CTL result, but displayed this data in Table 1 only.

Regarding the reviewer's final point on Figure 4b, the Table 1 differences stated are consistent with Figure 4b. The slope of HadEX2 in Figure 4b is 0.026 K/yr, the slope of CTL is 0.034 K/yr and the slope of IRR is 0.038 K/yr, which renders a bias of 0.008 K/yr for CTL and 0.012 K/yr for IRR, as detailed in Table 1. To ensure it is clear that Table 1 presents the bias and RMSE of the slopes (and not any other temperature parameter), the caption has been edited to state:

"Bias and Spatial RMSE of the Ensemble Mean Warming Trends (Slopes) of the CTL, IRR and CA Experiments Versus the Observational Products for the years 1981-2010a."

5. The introduction does not do a good job of introducing the main point of the paper. The first paragraph sets up the issue that observations show less warming in TXx than T2m, but models get it the other way around (TXx warms more than T2m). But we basically don't hear about this issue again. Subsequent paragraphs in the introduction are brief summaries of key papers (by the authors) and do not provide the cohesive overview of each topic a reader needs in an introduction, instead being based around a particular reference.

Reply. Thank you for highlighting this area for improvement. The introduction has now been substantially reworked to read:

[revised manuscript text omitted]

6. The abstract doesn't do a good job of summarizing the paper. It goes straight to "we did things" without explaining why the reader should be interested, or what the relevance is, then goes to "it's important" without presenting the evidence of why it is important. It provides no context for what was done, then the emphasis on the "pulse cooling" in the abstract is not followed through in the results. The subgrid scale aspect seems to be the most novel part of the paper, but since the rationale for it isn't explained clearly in the methods or results, it's a minority of the results section, there's no comparison for the scale of the water vapor feedbacks, and no clear explanation of why the water vapor feedbacks are responsible for the differences between the grid scale and subgrid scale, it's not convincing.

Reply. Thank you for this suggestion, we now reworked the abstract. Due to the addition of new data on the latent and sensible heat fluxes (see response to reviewer's point 9 below), we now clarify in the abstract that an increase in the LHF is responsible for the differences between the grid scale and subgrid scale. That is, at the land surface, the positive radiative forcing signal is too weak to offset the local cooling from the irrigation-induced increase in the latent heat flux. The updated abstract is detailed under our response to the reviewer's Point 1. Also regarding this comment, in our abstract, introduction and results section, a rationale for the subgrid scale aspect is now provided. Specifically, because we are looking at irrigation or CA-induced impacts at

the land surface, it is important to understand and quantify the effects of land management as such on local climate in order to distinguish between the effects of land management and other large-scale forcings such as a doubling of CO2.

7. The results section has paragraphs where the numbers in Tables are repeated, (with some unexplained deviations, as discussed above) with little attempt to give a view of what the results mean for the model's performance. There is no point in having a table if the text repeats what it says, and vice versa. There is absolutely a place for straightforward analysis and simple statements, but it needs to enhance clarity, not just be a list.

Reply. The section where we feel this point was most relevant has been substantially condensed (i.e. manuscript Section 3.2), to read:

"Neither irrigation nor CA has a cooling effect on T2m and TXx warming rates in irrigated/CA or non-irrigated/CA regions (Figure 3 and Table 2). The results suggest a slight irrigation- and CA-induced acceleration of the annual T2m and TXx warming trends, rather than the hypothesised cooling. For instance, irrigation induced an increased T2m warming rate of 0.0023 K yr-1 on average over land and 0.004 K yr-1 across all irrigated pixels. To put these increases into context, the mean T2m CRU observed warming trend over irrigated pixels was 0.029 K yr-1."

In addition, the text providing the ranges in Section 3.1 has been written more concisely, as described above under the reviewer's Point 4.

8. The Tables and Figures all need more detail in the captions, to help explain what they are and why they differ (and why those differences were deemed necessary). For instance: Table 1 – which years go towards the values? Table 2 – what are the "impact of irrigation and CA on various climatological values", because 0.026 K yr-1 for T2m doesn't make any sense as a number for the Control if it's supposed to be IRR – CTL as described in the caption. Figure 4 – presumably when it says "average" it's the mean, but then line 378 says "median", which is a notably different average.

Reply. Thank you for this suggestion. The caption for Table 1 has been updated to resolve this point as well as the reviewer's point 4, to read:

"Bias and Spatial RMSE of the Ensemble Mean Warming Trends (Slopes) of the CTL, IRR and CA Experiments Versus the Observational Products for the years 1981-2010a."

The caption for Table 2 has been updated to read:

"Impact of Irrigation and CA on Various Climatological Values (Absolute Slope Differences Calculated as IRR Minus CTL and CA Minus CTL for Grid-Scale, IRRSUB Minus RAIN and CASUB Minus CM for Subgrid-Scale) for the years 1981-2010a."

Regarding the spatial average (Figure 4 and 7), these figures show the mean temperature for all the pixels within each mask specified (i.e. all land, irrigated pixels or CA pixels) – plotted on the y-axis, for each year. The slope detailed in the figures was estimated using Sen's slope – so that all slope data used in this study is attained in the same way. As Sen's estimator takes the median slope (it was chosen as the nonparametric alternative to linear regression so that the slope is less sensitive to temperature outliers), the study conclusion thus notes (line 378):

"Insight into how modelled temperature is affected in its median by irrigation and CA over time was provided."

The captions for Figures 4 and 7 have been updated to clarify this. Figure 4 caption is: Spatial average of the warming rates for T2m (a, c and e) and TXx (b, d and f) for the CESM ensembles and observations. Data points specify the mean T2m and TXx temperatures for irrigated pixels (a-b), CA pixels (c-d), and (e-f) all land pixels. The slope was estimated using Sen's slope for the CTL (red), IRR (blue), CA (cyan), CRU (purple), HadEX2 (yellow), and GHCNDEX (black) temperatures.

The new caption for Figure 7 reads:

"Average of the TS warming rates over (a) irrigated pixels for the irrigated and rainfed

crop tiles; (b) CA pixels for the CA and CM crop tiles; (c) all pixels for the irrigated and rainfed crop tiles; and (d) all pixels for the CA and CM crop tiles. Spatial average of the SHF (e) and LHF (f) warming rates for the irrigated and rainfed crop tiles over irrigated pixels. Data points specify the mean TS, LHF and SHF values within the crop tiles and pixels specified. The slope was estimated using Sen's slope for the rainfed/CM (red), irrigated/CA (blue) experiments. For (a), (b), (e) and (f) the regions where less than 50% of the land pixels did not contain a value were excluded. For all land pixels (c and d), the minimum number of land pixels that needed to contain a value in order to be retained in the analysis was 15%."

In addition, the caption for Figure 1 has been updated to include reasons why the boxes differ and help explain better the distinctions, to read:

"Figure 1. (a) Percentage of each grid cell equipped for irrigation (%) (Siebert et al., 2005). (b) Potential estimate of CA extent mapped to the CLM crop PFT (Prestele et al., 2018). The red boxes in (a) denote the regional domains where irrigation is extensive and were thus examined in greater detail including Western North America (WNA), Central North America (CNA), south Europe and Mediterranean (MED), West Asia (WAS), South Asia (SAS), Southeast Asia (SEA), and East Asia (EAS). The red boxes in (b) denote the regional domains where CA is extensive and were thus examined in greater detail including WNA, CNA, MED, South-eastern South America (SSA), Central Europe (CEU) and Southern Australia (SAU)."

Figure 2 has been edited to read:

"Figure 2. Added value of including irrigation and CA in the simulated warming trends over 1981-2010. Absolute change in spatial root-mean-square error (RMSE) for the (a) IRR and (b) CA ensemble relative to the CTL ensemble over different regions (x axis) and with respect to 3 observational products (y axis). Considered regions are the SREX regions where irrigation is extensive (as highlighted in Figure 1a) and where CA is extensive (Figure 1b), in addition to global land, global irrigated land and global CA

land. Observational products are for near-surface air temperature T2m (CRU), annual maximum daytime temperature TXx (GHCNDEX and HadEX2). The spatial RMSEs are computed for the ensemble mean warming trend in every pixel, and subsequently averaged over the selected region. Regions with an observational coverage below 50% are marked in white."

And Figure 6 now reads:

"Figure 6. Subgrid-scale differences between the irrigated and rainfed crop tile in the IRR ensemble (irrigated minus rainfed) (a, g, k and o) and between CA and conventionally managed (CM) crops (CA minus CM) (e, h, l and p). For Ts (a-b), ET (g-h), LHF (k-l) and SHF (o-p). Grid-scale differences between the CTL and IRR ensemble (IRR minus CTL) (c, e, l, m and q) and between CA and conventionally managed (CM) crops (CA minus CM) (b, f, h, n and r). For Ts (c-d), ET (i-j), TMQ (e-f), LHF (m-n) and SHF (q-r), displayed over irrigated/CA pixels for comparative purposes. Differences are based on the ensemble mean warming trends of each experiment for 1981–2010. Hatching denotes less than 10% change induced by the model on mean warming trends of lumped ensemble members."

9. The results section has times where it would benefit from showing more evidence. For instance, the two paragraphs starting line 228 speculate that latent and sensible heat partitioning and changes in ET are responsible for the differences between CA and IRR, and the Control. But instead of exploring these and showing how the latent heat changes, there is just references to previous papers. I.e. it is not a result, it is a summary of previously published research. Similarly, the paragraphs line 289 – 315 contain a lot of speculation and references and not enough evidence.

Reply. New sets of results have been added that provide evidence on the changes in the latent heat flux (LHF) and sensible heat flux (SHF). Trend data on the grid- and subgrid-scale LHF and SHF has been provided in Table 2. Figure 6 now shows a comparison between the subgrid and grid scale LHF and SHF changes for irrigated

and CA land, while the new Figure 7 details the spatial average of LHF and SHF on irrigated pixels (for the figure additions, please see above under the Reviewer's Point 1). This has also helped to address the reviewer's Point 10 below regarding the discussion section below and the inclusion of a reflection on uncertainties in the partitioning of latent and sensible heat when albedo changes.

10. The discussion section is missing, at the very least: cloud uncertainties (different models do them differently, and they are notoriously difficult to resolve, so that these results rely on them is problematic); uncertainties in the partitioning of latent and sensible heat when albedo changes (i.e. Bowen ratio; the fact that the CA increase in albedo is a huge assumption, as soil albedo is very heterogeneous and dependent partly on soil moisture (thus the CA modeled might be doing the wrong thing in many areas); the representation of transpiration in the model, and the fact that presumably the crop/vegetation cover is the same when in reality these changes would affect the LAI of the crop; the canopy interception and soil interception representation in the model, which affects the evaporation and thus how much the irrigation and CA affect the evaporation.

Reply. The discussion sections has been rewritten to address these points as follows:

[revised manuscript text omitted]

Thank you for your time and effort in helping to improve our paper. It is great appreciated.

Please also note the supplement to this comment:
https://esd.copernicus.org/preprints/esd-2020-35/esd-2020-35-AC1-supplement.pdf

[Figure]

*Response Figure 1 (adding to Figure 6 in the paper). Subgrid-scale differences between the irrigated and rainfed crop tile in the IRR ensemble (irrigated minus rainfed) (a, g, k and o) and between CA and conventionally managed (CM) crops (CA minus CM) (e, h, l and p). For $T_s$ (a-b), ET (g-h), LHF (k-l) and SHF (o-p). Grid-scale differences between the CTL and IRR ensemble (IRR minus CTL) (c, e, l, m and q) and between CA and conventionally managed (CM) crops (CA minus CM) (b, f, h, n and r). For $T_s$ (c-d), ET (i-j), TMQ (e-f), LHF (m-n) and SHF (q-r), displayed over irrigated/CA pixels for comparative purposes. Differences are based on the ensemble mean warming trends of each experiment for 1981–2010. Hatching denotes less than 10% change induced by the model on mean warming trends of lumped ensemble members.*

**Fig. 1.** Response Figure 1 (adding to Figure 6 in the paper).

[Figure]

*Response Figure 2 (added to Figure 7 in the paper). Spatial average of the SHF (left) and LHF (right) warming rates for the irrigated and rainfed crop tiles over irrigated pixels. Data points specify the mean LHF and SHF values within the crop tiles and pixels specified. The slope was estimated using Sen's slope for the rainfed/CM (red), irrigated/CA (blue) experiments for the years 1981-2010.*

**Fig. 2.** Response Figure 2 (added to Figure 7 in the paper).
Interactive
comment

[Figure]

*Response Figure 3 (replacing the paper's Figure 2). Added value of including irrigation and CA in the simulated warming trends over 1981-2010. Absolute change in spatial root-mean-square error (RMSE) for the (a) IRR and (b) CA ensemble relative to the CTL ensemble over different regions (x axis) and with respect to 3 observational products (y axis). Considered regions are the SREX regions where irrigation is extensive (as highlighted in Figure 1a) and where CA is extensive (Figure 1b), in addition to global land, global irrigated land and global CA land. Observational products are for near-surface air temperature $T_{2m}$ (CRU), annual maximum daytime temperature TXx (GHCNDEX and HadEX2). The spatial RMSEs are computed for the ensemble mean warming trend in every pixel, and subsequently averaged over the selected region. Regions with an observational coverage below 50% are marked in white.*

**Fig. 3.** Response Figure 3 (replacing the paper's Figure 2).

---

## Author Comment (AC3) · 19 Feb 2021

Note that the attached table is complimentary to our response to the reviewer's 2 point and is added to the revised paper as Table 3. It details the bias and spatial RMSE of the ensemble mean warming trends (slopes) of the RAIN, IRRSUB, CASUB and CM experiments Versus the E-OBS (K yr-1) observational Product for the years 1981-2010.

[Figure]

**Response Table 1** *(added as Table 3 in the paper). Bias and Spatial RMSE of the Ensemble Mean Warming Trends (Slopes) of the RAIN, IRR$_{SUB}$, CA$_{SUB}$ and CM Experiments Versus the E-OBS (K yr$^{-1}$) Observational Product for the years 1981-2010.*

| Irrigated MED pixels bias | | CA MED pixels bias | | Irrigated MED pixels RMSE | | CA MED pixels RMSE | |
|---|---|---|---|---|---|---|---|
| RAIN | IRR$_{SUB}$ | CM | CA$_{SUB}$ | RAIN | IRR$_{SUB}$ | CM | CA$_{SUB}$ |
| 0.015 | 0.004 | 0.013 | 0.022 | 0.028 | 0.031 | 0.027 | 0.026 |

**Fig. 1.** Response Table 1 (added as Table 3 in the paper).

[Figure]

---

## Author Response (AR1)

This paper looks at the effects of prescribed representations of conservation agriculture and irrigation on mean annual 2m and maximum daytime temperature in CESM. There is some interesting analysis and potential for results that could be useful for the community. There are aspects of the sub-grid scale vs grid scale analysis, and possibility for critique of whether more processes enhance model skill, that are intriguing. The figures are generally well presented. However, there are several issues that need to be addressed.

We greatly thank the reviewer for the appreciation of the manuscript and for the constructive comments, which greatly helped to improve the quality of the study. Here below, we provide a point-by-point response to each comment. The modified manuscript text is shown in italics.

1.  The paper reads like a combination of previously published results (specifically, the ensembles used are already published in Theiry et al. (2017) and Hirsch et al. (2018)). That might be unfair, but the regression analyses is simple and it seems unlikely it wasn't done separately for CA and irrigation, and much of the explanatory analysis references these two papers. It is the responsibility of the authors to show clearly why this is novel compared to what has come before.

**Reply.** We confirm that the analysis presented here is based on simulations that have been published previously. However, we believe that this new study moves beyond the state of the art in three ways. First, the main novelty of the current study lies in the explicit focus on trends, whereas previous studies focused on the influence of land management on the climatology (of means and extreme indicators). Second, the explicit focus on the subgrid versus grid-scale response offers important new insights on the local land surface land surface response to land management. Third, this study for the first time explicitly considers the radiative forcing resulting from realistic land management. We find that the positive radiative forcing signal arising from enhanced atmospheric water vapour is too weak to offset the local cooling from the irrigation-induced increase in the latent heat flux. This is been emphasised more clearly by including new evidence on the latent and sensible heat fluxes (warming/cooling trends as well as spatial averages) and the results and abstract have been adjusted to reflect these results as well as the reviewer's Point 2 and Point 6.

The new abstract reads as:

> *Abstract. Understanding and quantifying land management impacts on local climate is important for distinguishing between the effects of land management and large-scale climate forcings. This study for the first time explicitly considers the radiative forcing resulting from realistic land management and offers new insights on the local land surface response to land management. Regression-based trend analysis is applied to observations and present-day ensemble simulations with the Community Earth System Model (CESM) version 1.2.2 to assess the impact of irrigation and conservation agriculture (CA) on warming trends using an approach that is less sensitive to temperature extremes. At the regional scale, an irrigation- and CA-induced acceleration of the annual mean near-surface air temperature ($T_{2m}$) warming trends and the annual maximum daytime temperature (TXx) warming trends were evident. Estimation of the impact of irrigation and CA on the spatial average of the warming trends indicated that irrigation and CA have a pulse cooling effect on $T_{2m}$ and TXx, after which the warming trends increase at a greater rate than the control simulations. This differed at the local (subgrid) scale under irrigation where surface temperature cooling and the dampening of warming trends were both evident. As the local surface warming trends, in contrast to regional trends, do not account for atmospheric (water vapour) feedbacks, their dampening confirms the importance of atmospheric feedbacks (water vapour forcing) in explaining the enhanced regional trends. At the land surface, the positive radiative forcing signal arising from enhanced atmospheric water vapour is too weak to offset the local cooling from the irrigation-induced increase in the evaporative fraction. Our results underline that agricultural management has complex and nonnegligible impacts on the local climate and highlight the need to carefully represent and evaluate land management in climate models.*

The new evidence, which has been added to the paper's Figure 6, 7 and 8, includes the subgrid-scale differences between the irrigated and rainfed crop tile in the IRR ensemble and between CA and conventionally managed (CM) crops, for the latent heat flux (LHF) (Response Figure 1a and 1b below) and the sensible heat flux (SHF) (1c and 1d below). Grid-scale differences between the CTL and IRR ensemble and between CA and CM crops for LHF (Response Figure 2a and 2b) and SHF (2b and 2c) are also included over irrigated/CA pixels for comparative purposes, as detailed below. In addition, to the paper's Figure 8, data on the spatial average of the SHF and LHF warming rates for the irrigated and rainfed crop tiles over irrigated pixels, is now included, as shown in our Response Figure 3.

[Figure]

***Response Figure 1*** *(added to Figure 7 of the paper). Subgrid-scale differences between the irrigated and rainfed crop tile in the IRR ensemble (irrigated minus rainfed) (a and c) and between CA and conventionally managed (CM) crops (CA minus CM) (b and). For LHF (a-b) and SHF (c-d), displayed over irrigated/CA pixels for comparative purposes. Differences are based on the ensemble mean warming trends of each experiment for 1981–2010. Hatching denotes less than 10% change induced by the model on mean warming trends of lumped ensemble members.*

[Figure]

*Response Figure 2 (added to Figure 6 in the paper). Grid-scale differences between the CTL and IRR ensemble (IRR minus CTL) (a and c) and between the CTL and CA ensemble (CA minus CTL) (b and dj). For LHF (a-b) and SHF (c-d), displayed over irrigated/CA pixels for comparative purposes. Differences are based on the ensemble mean warming trends of each experiment for 1981–2010. Hatching denotes less than 10% change induced by the model on mean warming trends of lumped ensemble members.*

[Figure]

*Response Figure 3 (added as Figure 8g and 8h in the paper). Spatial average of the SHF (left) and LHF (right) warming rates for the irrigated and rainfed crop tiles over irrigated pixels. Data points specify the mean LHF and SHF values within the crop tiles and pixels specified. The slope was estimated using Sen's slope for the rainfed/CM (red), irrigated/CA (blue) experiments for the years 1981-2010.*

2.  The results (as shown in Table 1 especially) are difficult to reconcile with the statements made in the abstract and conclusions. Looking at Table 1, if the smallest RMSE (or the anomalies closest to zero) are considered, the Control simulation is better ~ 2/3 of the time. The abstract says, "our results underline. . . the need to account for land management in climate projections". Surely the opposite is true, as the Control scenario does better by the measure most used to assess model skill. Even within the results, there appear to be contradictions. Line 218: "the impact of irrigation and CA on the modeled spatially averaged temperatures. . .

is an overall cooling effect". Line 223: "for the IRR and CA models. . . the spatially averaged T2m and TXx warming rates are higher than those of the CTL model".

**Reply.** The CTL simulation is better when considering the extreme (TXx) temperature results but not when considering mean temperatures (T2m). Particularly in the case of T2m for irrigation, the results in Table 1 show all cases are better all of the time. In one case for CA (all land), the RMSEs are equivalent, but otherwise the CTL RMSE is higher. So we disagree that the opposite is true in general, but we do agree that the statement in the abstract can be refined. The adjusted abstract statement reads as follows:

*Our results underline that agricultural management has complex and nonnegligible impacts on the local climate and highlight the need to carefully represent and evaluate land management in climate models.*

Regarding the apparent contradiction between lines 218 and 223, the 'cooling effect' noted in line 218 (Figure 4) refers to a decrease in absolute temperature, that is, the intercept of the regression. Line 223, on the other hand, does refer to the trend over time, that is, the slope of the regression. This distinction has now been clarified in the text, which now reads:

*However, the impact of irrigation and CA on the modelled spatially averaged temperatures improves the closeness to that of the observations, i.e. there is an overall decrease in absolute temperature (Figure 4a-d), which is consistent with current theory (Kueppers et al., 2007; Saeed et al., 2009; Kueppers and Snyder, 2012; Thiery et al., 2017, 2020; Hirsch et al., 2018).*

3.   Some of the results are presented in such a way as to be somewhat misleading. For instance, the values in Figure 2 (% change in RMSE) with the colored categorization (which, being visual, is much stronger evidence to the reader than a table) can be compared with the equivalent anomaly in Table 1 (RMSE). For irrigation (IRR-CTL) the RMSE T2m (CRU) difference is -0.002, and the figure categorization is - 5-10%. For irrigation (IRR-CTL) the RMSE TXx GHCNDEX difference is +0.004, and the figure categorization is 0-5%. i.e. a difference in RMSE that is twice as big, is categorised as half the size in terms of color. This means that it looks as though the CA and IRR simulations are doing much better than if the simple RMSE is considered.

**Reply.** We agree with this point. In the revised figure 2, we now present the absolute change in the RMSE data (in K), as detailed in our Response Figure 3.

[Figure]

*Response Figure 3 (replacing the paper's Figure 2). Added value of including irrigation and CA in the simulated warming trends over 1981-2010. Absolute change in spatial root-mean-square error (RMSE) for the (a) IRR and (b) CA ensemble relative to the CTL ensemble over different regions (x axis) and with respect to 3 observational products (y axis). Considered regions are the SREX regions where irrigation is extensive (as highlighted in Figure 1a) and where CA is extensive (Figure 1b), in addition to global land, global irrigated land and global CA land. Observational products are for near-surface air temperature $T_{2m}$ (CRU), annual maximum daytime temperature TXx (GHCNDEX and HadEX2). The spatial RMSEs are computed for the ensemble mean warming trend in every pixel, and subsequently averaged over the selected region. Regions with an observational coverage below 50% are marked in white.*

**4.**   Some of the results are inconsistent with each other. For instance, on line 182, the range of temperature anomaly compared to observations is given as 0.007 - 0.03 in the text, but in Table 1 it is 0.007 – 0.024 (usually a number is rounded down when the last value is below 5). Or line 179 where the text says 0.004 for the Control, but Table 1 says 0.006. Figure 4b shows TXx HadEX2 on top of IRR and CNT much higher, but Table 1 shows CTL and IRR with differences from HadEX2 of 0.008 and 0.012 respectively.

**Reply.** The 0.007 - 0.03 in the text refers to the range of temperature anomaly for all three (CTL, IRR and CA) experiments, not just the CTL, which is consistent with the data in Table 1 (the 0.03 K yr$^{-1}$ TXx bias is noted for the GHCNDEX observations and the IRR ensemble). This has now been clarified in the text, as follows:

*On average, the CTL, IRR and CA ensembles overestimate TXx warming trends by ~0.007–0.03 K yr$^{-1}$ over all land pixels. Over irrigated pixels, the CTL and IRR ensemble overestimate TXx by ~0.008–0.013 K yr$^{-1}$. Over CA pixels, the CTL and CA ensemble overestimate TXx by ~0.006–0.013 K yr$^{-1}$.*

For the reviewer's point regarding line 179, indeed the text referring to CTL T2m bias should read according to the data in Table 1 – i.e. 0.006. In order to also address the reviewer's point 7 below, we have addressed this by not including the statement in the text that specifies the CTL result, but displayed this data in Table 1 only.

Regarding the reviewer's final point on Figure 4b, the Table 1 differences stated are consistent with Figure 4b. The slope of HadEX2 in Figure 4b is 0.026 K/yr, the slope of CTL is 0.034 K/yr and the slope of IRR is 0.038 K/yr, which renders a bias of 0.008 K/yr for CTL and 0.012 K/yr for IRR, as detailed in Table 1. To ensure it is clear that Table 1 presents the bias and RMSE of the slopes (and not any other temperature parameter), the caption has been edited to state:

> *Bias and spatial RMSE of the ensemble mean warming trends (slopes) of the CTL, IRR and CA experiments versus the observational products for the years 1981-2010[a].*

5. The introduction does not do a good job of introducing the main point of the paper. The first paragraph sets up the issue that observations show less warming in TXx than T2m, but models get it the other way around (TXx warms more than T2m). But we basically don't hear about this issue again. Subsequent paragraphs in the introduction are brief summaries of key papers (by the authors) and do not provide the cohesive overview of each topic a reader needs in an introduction, instead being based around a particular reference.

**Reply.** Thank you for highlighting this area for improvement. The introduction has now been substantially reworked to improve the storyline, update the cited studies, include a discussion on the relevance of GCMs when studying LULC effects on climate, motivate the subgrid perspective, and highlight the novelty of the study. The revised introduction now reads:

[revised manuscript text omitted]

*Considering the potential effects of irrigation and CA on climate (Thiery et al., 2017), it is possible that the discrepancies between climate models and observations regarding temperature changes (Donat et al., 2017) are because the models exclude the effect of agricultural management techniques on temperature. The goal of this study is thus to test the hypothesis that CESM version 1.2.2 overestimates warming trends in some regions because irrigation and CA are excluded. That is, warming rates are hypothesized to increase at a slower rate – showing signs of cooling, in irrigation- and CA-affected regions when climate models do account for a theoretical constant level of these land management practices. To realise this goal, the following objectives were formulated: (1) determine spatial warming rates using simulations that account for irrigation and CA and inspect whether CESM overestimates warming trends; (2) compare the observed rates of warming to the modelled rates of warming for irrigated and CA pixels, as well as non-irrigated and non-CA pixels; and (3) estimate the impact of irrigation on the spatial average of the warming rates over time for all land, selected regions, and irrigated and CA pixels. Within this framework, the novelty of this study lies in (i) an explicit focus on land management impacts on trends as opposed to the climatology; (ii) a comparison of the subgrid versus grid-scale response, offering important new insights on the local land surface land surface response to land management; and (iii) consideration of the radiative forcing resulting from realistic land management.*

6. The abstract doesn't do a good job of summarizing the paper. It goes straight to "we did things" without explaining why the reader should be interested, or what the relevance is, then goes to "it's important" without presenting the evidence of why it is important. It provides no context for what was done, then the emphasis on the "pulse cooling" in the abstract is not followed through in the results. The subgrid scale aspect seems to be the most novel part of the paper, but since the rationale for it isn't explained clearly in the methods or results, it's a minority of the results section, there's no comparison for the scale of the water vapor feedbacks, and no clear explanation of why the water vapor feedbacks are responsible for the differences between the grid scale and subgrid scale, it's not convincing.

**Reply.** Thank you for this suggestion, we now reworked the abstract. Due to the addition of new data on the latent and sensible heat fluxes (see response to reviewer's point 9 below), we now clarify in the abstract that an increase in the LHF is responsible for the differences between the grid scale and subgrid scale. That is, at the land surface, the positive radiative forcing signal originating from the enhanced atmospheric water vapour is too weak to offset the local cooling from the irrigation-induced increase in the latent heat flux. The updated abstract is detailed under our response to the reviewer's Point 1. Also regarding this comment, in our abstract, introduction and results section, a rationale for the subgrid scale aspect is now provided. Specifically, because we are looking at irrigation or CA-induced impacts at the land surface, It is important to understand and quantify the effects of land management as such on local climate in order to distinguish between the effects of land management and other large-scale forcings such as a doubling of $CO_2$.

7. The results section has paragraphs where the numbers in Tables are repeated, (with some unexplained deviations, as discussed above) with little attempt to give a view of what the results mean for the model's performance. There is no point in having a table if the text repeats what it says, and vice versa. There is absolutely a place for straightforward analysis and simple statements, but it needs to enhance clarity, not just be a list.

**Reply.** The section where we feel this point was most relevant has been substantially condensed (i.e. manuscript Section 3.2), to read:

*Neither irrigation nor CA has a cooling effect on $T_{2m}$ and TXx warming rates in irrigated/CA or non-irrigated/CA regions (Figure 3 and Table 2). The results suggest a slight irrigation- and CA-induced acceleration of the annual $T_{2m}$ and TXx warming trends, rather than the hypothesised cooling. For instance, irrigation induced an increased $T_{2m}$ warming rate of 0.0023 K yr$^{-1}$ on average over land and 0.004 K yr$^{-1}$ across all irrigated pixels. To put these increases into context, the mean $T_{2m}$ CRU observed warming trend over irrigated pixels was 0.029 K yr$^{-1}$.*

In addition, the text providing the ranges in Section 3.1 has been written more concisely, as described above under the reviewer's Point 4.

8. The Tables and Figures all need more detail in the captions, to help explain what they are and why they differ (and why those differences were deemed necessary). For instance: Table 1 – which years go towards the values? Table 2 – what are the "impact of irrigation and CA on various climatological values", because 0.026 K yr-1 for T2m doesn't make any sense as a number for the Control if it's supposed to be IRR – CTL as described in the caption. Figure 4 – presumably when it says "average" it's the mean, but then line 378 says "median", which is a notably different average.

**Reply.** Thank you for this suggestion. The caption for Table 1 has been updated to resolve this point as well as the Reviewer's point 4, to read:

*Bias and spatial RMSE of the ensemble mean warming trends (slopes) of the CTL, IRR and CA experiments versus the observational products for the years 1981-2010.*

The caption for Table 2 (now Table 3 in the paper) has been updated to read:

*Impact of irrigation and CA on various climatological values (absolute slope differences calculated as IRR minus CTL and CA minus CTL for grid-scale, IRR$_{SUB}$ minus RAIN and CA$_{SUB}$ minus CM for subgrid-scale) for the years 1981-2010.*

Regarding the spatial average (Figure 4 and 7), these figures show the mean temperature for all the pixels within each mask specified (i.e. all land, irrigated pixels or CA pixels) – plotted on the y-axis, for each year. The slope detailed in the figures was estimated using Sen's slope – so that all slope data used in this study is attained in the same way. As Sen's estimator takes the median slope (it was chosen as the nonparametric alternative to linear regression so that the slope is less sensitive to temperature outliers), the study conclusion thus notes (line 378):

*Insight into how modelled temperature is affected in its median by irrigation and CA over time was provided.*

The captions for Figures 4 and 7 (Figure 4 and Figure 8 in the revised paper) have been updated to clarify this. Figure 4's caption is:

[revised manuscript text omitted]

9. The results section has times where it would benefit from showing more evidence. For instance, the two paragraphs starting line 228 speculate that latent and sensible heat partitioning and changes in ET are responsible for the differences between CA and IRR, and the Control. But instead of exploring these and showing how the latent heat changes, there is just references to previous papers. I.e. it is not a result, it is a summary of previously published research. Similarly, the paragraphs line 289 – 315 contain a lot of speculation and references and not enough evidence.

**Reply.** New sets of results have been added that provide evidence on the changes in the latent heat flux (LHF) and sensible heat flux (SHF). Trend data on the grid- and subgrid-scale LHF and SHF has been provided in Table 3. Figure 6 and Figure 7 now allow for a comparison between the subgrid and grid scale LHF and SHF changes for irrigated and CA land, while the new Figure 8 details the spatial average of LHF and SHF on irrigated pixels (for the figure additions, please see above under the Reviewer's Point 1). This has also helped to address the reviewer's Point 10 below regarding the discussion section below and the inclusion of a reflection on uncertainties in the partitioning of latent and sensible heat when albedo changes.

10. The discussion section is missing, at the very least: cloud uncertainties (different models do them differently, and they are notoriously difficult to resolve, so that these results rely on them is problematic); uncertainties in the partitioning of latent and sensible heat when albedo changes (i.e. Bowen ratio; the fact that the CA increase in albedo is a huge assumption, as soil albedo is very heterogeneous and dependent partly on soil moisture (thus the CA modeled might be doing the wrong thing in many areas); the representation of transpiration in the model, and the fact that presumably the crop/vegetation cover is the same when in reality these changes would affect the LAI of the crop; the canopy interception and soil interception representation in the model, which affects the evaporation and thus how much the irrigation and CA affect the evaporation.

**Reply.** Paragraphs in the discussion section have been rewritten to address these points as follows:

*Paragraph 3:*
*Although this study was constructed with great care and built on a state-of-the-art modelling suite, several future developments could improve understanding of the impact of irrigation and CA on climate. Firstly, the quality of the model(s) could be improved by using transient irrigation and CA extents and new land cover datasets from the 6th phase of the Coupled Model Intercomparison Project (CMIP6) (Lawrence et al., 2016). In this study, a static irrigation map for the year 2000 was used for the whole simulation period. This likely contributes to our results being conservative. If, for instance, irrigation expands over time, the cooling effect may become stronger and thus affect the warming trends. Furthermore, the extent to which the increase in surface albedo (i.e. the first competing effect of CA) affects the sensible and latent heat fluxes partly depends on soil moisture, which too is not static. Also, CMIP6 experiments are based on annual emissions, whereas CMIP5 was based on decadal emissions and CMIP6 models were updated with irrigation-related features and land cover maps that incorporate irrigation and CA expansion over time (Goddard et al., 2013; Miao et al., 2014; Boer et al., 2016; Meinshausen et al., 2017; Stouffer et al., 2017). CMIP6 models may therefore improve the dynamics between irrigation, CA and climate change, provided that they represent these land management techniques in their surface schemes.*

*Paragraph 4:*
*The second consideration is that all simulations used in this study (5 control, 5 irrigation and 5 CA) were from a single model. Ensembles completed as such with the same model but different simulations (i.e. based on different initial conditions) characterise the uncertainty associated with internal climate variability only, while multi-model ensembles also account for the impact of model differences (Tebaldi and Knutti, 2007; Knutti et al., 2010). This limitation can impact cloud uncertainties. Hirsch et al. (2017) found that the CESM tends to produce large cloud feedbacks over Central Europe, Central North America, North Asia, and South Asia when more energy is reflected at the surface. Irrigation-induced increases in latent heat fluxes led to more water vapor in the lower atmosphere, which generated low-level clouds (see also Sherwood et al., 2017). This limited shortwave radiation and hence the amount of energy available at the surface because the increased cloud cover reflected more downward shortwave radiation above the cloud layer, resulting in surface cooling. This was enhanced by a corresponding decrease in sensible heat fluxes, reflecting the decrease in the amount of energy available at the surface and/or the increase in latent heating. The impact of cloud cover combined with land management change remains challenging to resolve. Therefore, this study should ideally be repeated with other models. Donat et al. (2017), for instance, conducted their study on 20 CMIP5 models, but these models did not incorporate irrigation and CA.*

*Paragraph 6:*
*The final consideration is whether regression-based models are suitable for analysing changes in highly variable climate data, particularly annual extreme temperature data (von Storch, 2006). Essentially, the regression slope blends forced temperature change and variability, to provide an estimation of the temperature variation over time – within which variance can be lost due to noisy data. Whether the TXx and $T_{2m}$ temperatures were first spatially averaged and then the slope retrieved or if each slope was estimated for each pixel and then the overall trends examined, the outcome remains. This is unsurprising considering that in the spatial averaging the noise contributions are averaged out, while the individual regression data suffers from the variance loss related to regression. However, when applied to over 60 years of observational data, the regression model used in this study showed similar trends to using the difference between the past and the present average temperatures (not shown). This implies that the irrigation and CA-inclusive climate system may require a longer timeframe (than the 30 years plus a 5-year spin-up period used) for trends to overtake the natural variability. Additionally, rather than aggregating all months, trends during individual months or seasons could be examined. This can affect, for instance, the influence of irrigation on $T_s$, which has a clear seasonal pattern, with more cooling during the driest and/or hottest months (Thiery et al., 2017). A smaller magnitude in TXx response to CA at the subgrid-scale has also been noted during the summer season due to a larger leaf area index (LAI) reducing soil surface exposure and thus the contrast between CA and conventionally managed crops (Hirsch et al., 2018). Furthermore, the implementation of CA within CESM does not capture crop planting and harvesting cycling (Davin et al., 2014), which would affect the LAI of the crop and potentially the effect of CA on surface climate.*

Thank you for your time and effort in helping to improve our paper. It is great appreciated.

The authors' summary statement in the abstract is certainly an informative conclusion. They write
 *"Our results underline that agricultural management has complex and nonnegligible impacts on the local climate and highlights the need to account for land management in climate projections."*

And further that
 *"It remains challenging to resolve this, however, because it is difficult to separate land management from other effects in GCMs – particularly natural climate variability (Cook et al., 2015)".*

The Reviewer's time is greatly appreciated, and we believe that by addressing the Reviewer's comments as outlined below, it has enhanced the value and quality of the paper. A point-by-point response to each comment is detailed below and the amended manuscript text is provided in italics.

1. They summarize their paper with the text
    *"The goal of this study is thus to test the hypothesis that CESM version 1.2.2 overestimates warming trends in some regions because irrigation and CA are excluded. That is, warming rates are hypothesised to decline – showing signs of cooling, in irrigation- and CA-affected regions when climate models do account for a theoretical constant level of these land management practices. To realise this goal, the following three objectives were formulated: (1) Determine spatial warming rates using GCM simulations that account for irrigation and CA and inspect whether CESM overestimates warming trends; (2) Compare the observed rates of warming to the modelled rates of warming for irrigated and CA pixels, as well as nonirrigated and non-CA pixels; and (3) Estimate the impact of irrigation on the spatial average of the warming rates over time (1981-2010) for all land, selected regions, and irrigated and CA pixels."*

   However, the basis to quantify these impacts is flawed, or at least significantly muddled. First, model comparison studies are just model sensitivity studies. Without an assessment of model skill with the appropriate real world observed data, this is an incomplete (and potentially misleading) approach. The real world data needs to be on the spatial and temporal scale of the effect they are assessing (irrigation and conservation agriculture). The recent GRAINEX project quantified these scales [https://www.eol.ucar.edu/field_projects/grainex]. The model results should be compared against such data.

   Indeed there are numerous regional, mesoscale and local studies that have assessed the role of irrigation and land management on weather and climate. The authors do not seem to be familiar with this research. Here are just a few

   Adegoke, J.O., R.A. Pielke Sr., J. Eastman, R. Mahmood, and K.G. Hubbard, 2003: Impact of irrigation on midsummer surface fluxes and temperature under dry synoptic conditions: A regional atmospheric model study of the U.S. High Plains. Mon. Wea. Rev., 131, 556-564.

   Betts RA. Implications of land ecosystem-atmosphere interactions for strategies for climate change adaptation and mitigation. Tellus B 2007, 59:602–615. doi:10.1111/j.1600-0889.2007.00284.x.
   Boyaj et al, 2020: Increasing heavy rainfall events in south India due to changing land use and land cover. QJRMS https://doi.org/10.1002/qj.3826.
   Chen, C. J., C. C. Chen, M. H. Lo, J. Y. Juang, and C. M. Chang, 2020: Central Taiwan's hydroclimate in response to land use/cover change. *Env. Res. Lett.,* **15,** 034015

   Douglas, E.M., D. Niyogi, S. Frolking, J.B. Yeluripati, R. A. Pielke Sr., N. Niyogi, C.J. Vörösmarty, and U.C. Mohanty, 2006: Changes in moisture and energy fluxes due to agricultural land use and irrigation in the Indian Monsoon Belt. Geophys. Res. Letts, 33, doi:10.1029/2006GL026550.

   He, Y., E. Lee, and J. S. Mankin, 2019: Seasonal tropospheric cooling in Northeast China associated with cropland expansion. Env. Res. Lett. 15, 034032.

   Hossain, F., J. Arnold, E. Beighley, C. Brown, S. Burian, J. Chen, S. Madadgar, A. Mitra, D. Niyogi, R.A. Pielke Sr., V. Tidwell, and D. Wegner, 2015: Local-to-regional landscape drivers of extreme weather and climate: Implications for water infrastructure resilience. J. Hydrol. Eng., **10.1061/(ASCE)HE.1943- 5584.0001210** , 02515002.

   Pielke Sr., R.A., 2001: Influence of the spatial distribution of vegetation and soils on the prediction of cumulus convective rainfall. Rev. Geophys., 39, 151-177.

   Pielke Sr., R.A., R. Mahmood, and C. McAlpine, 2016: Land's complex role in climate change. Physics Today, 69(11), 40.

   Ullah et al 2020: How Vegetation Spatially Alters the Response of Precipitation and Air Temperature? Evidence from Pakistan. Asian Journal of Atmospheric Environment 14(2): 133- 145.

   Woldemichael, A.T., F. Hossain, and R. A. Pielke Sr., 2014: Impacts of post-dam land-use/land-cover changes on modification of extreme precipitation in contrasting hydro-climate and terrain features. J. Hydrometeor., 15, 777–800, doi:10.1175/JHM-D-13-085.1.

   Zhang, T., R. Mahmood, X. Lin, and R.A. Pielke Sr., 2019: Irrigation impacts on minimum and maximum surface moist enthalpy in the Central Great Plains of the USA. Weather and Climate Extremes, 23, https://doi.org/10.1016/j.wace.2019.100197.

**Reply.** Thank you for raising these important points. We firstly wish to highlight that our work is not merely a model sensitivity study, as we compare our simulations against three observational products (see Figure 2 and Table 1 in the paper). Figure 2 has now been updated

(shown below as 'Response Figure 1') to display the absolute change in spatial root-mean-square error for the IRR and CA ensemble relative to the CTL ensemble over different regions and with respect to 3 observational products. The paper's Table 1 details the bias and spatial RMSE of the ensemble mean warming trends of the CTL, IRR and CA experiments versus the observational products. We do however agree that including additional observational products for comparison at the subgrid scale would improve the completeness of the approach. The products recommended by the Reviewer are appreciated, but given the global scope of our analysis as well as the focus on trends; we feel that the spatial and temporal coverage of this data is inadequate for validation of global model outputs. To resolve this and address the Reviewer's point, we have added an analysis of the E-OBS European CDG data. Please see our response to Reviewer point 3 below for full details regarding this additional analysis.

[Figure]

*Response Figure 1 (Figure 2 in the paper). Added value of including irrigation and CA in the simulated warming trends over 1981-2010. Absolute change in spatial root-mean-square error (RMSE) for the (a) IRR and (b) CA ensemble relative to the CTL ensemble over different regions (x axis) and with respect to 3 observational products (y axis). Considered regions are the SREX regions where irrigation is extensive (as highlighted in the paper's Figure 1a) and where CA is extensive (see Figure 1b of the paper), in addition to global land, global irrigated land and global CA land. Observational products are for near-surface air temperature $T_{2m}$ (CRU), annual maximum daytime temperature TXx (GHCNDEX and HadEX2). The spatial RMSEs are computed for the ensemble mean warming trend in every pixel, and subsequently averaged over the selected region. Regions with an observational coverage below 50% are marked in white.*

We also agree as to the benefits of including more local studies that have assessed the role of irrigation and land management on weather and climate. Thus, the introduction has been substantially reworked to now read as follows:

*"Conservation agriculture (CA), which involves crop residue management, crop rotation (Carrer et al., 2018; Lombardozzi et al., 2018) and minimal or no tillage (Kassam et al., 2015), can create climate feedbacks due to the presence of a crop residue over CA land that change both the radiative and hydrological properties at the surface (Davin et al., 2014). Hirsch et al. (2018) explored whether applying the no-till component of CA within the Community Earth System Model (CESM) improves the simulation of present-day climate. They found that the surface temperature response was influenced by three competing effects: (1) a surface albedo increase – which reduces the availability of energy for partitioning between the sensible and latent heat fluxes; (2) increased surface resistance (e.g. from mulch) – which reduces soil evaporation; and (3) increased soil moisture retention leading to enhanced transpiration. The local cooling response to CA was somewhat counteracted by grid-scale changes in climate over North America, Europe, and Asia because of negative atmospheric feedbacks. Grid-scale changes in climate counteracting local responses to land use change has also been demonstrated by Malyshev et al. (2015) who showed that the subgrid signal of land use change in near surface temperature was diminished by the averaging with undisturbed portions of the pixels. The importance of local-scale responses to land cover change has also been indicated in observation-based studies (e.g., Mahmood et al., 2014; Li et al., 2015), yet few global-scale modelling studies examine the local land surface response to land management (Paulot et al., 2018; Meier et al., 2018).*

*Using GCMs, such as CESM, to simulate land-atmosphere interactions for investigating the effects of irrigation and agricultural conversion has been criticized as insufficient (Niyogi et al., 2002). This is partly because their coarse resolution (e.g., of order 100 km) hampers their performances in describing the present-day climate at the regional scale (Jiang et al., 2016). Furthermore, economic, societal and water resource factors are ignored – a void that initiated the so-called 'bottom-up' approach to evaluating the effects of land-use change (Douglas et al., 2006). Regarding the applicability of the knowledge produced by GCMs, they do not provide the skill required at the spatial scale to offer practical responses at the infrastructure scale (Hossain et al., 2015). Despite these constraints, GCMs remain a prime tool for projecting changes in the climate system (Fajardo et al., 2020; Gupta et al., 2020; Hofer et al., 2020). Examples include the GCMs that are part of the latest Coupled Model Intercomparison Project (CMIP6) and used by the IPCC in consecutive assessment reports (Yazdandoost et al., 2021). However, these GCMs largely exclude agricultural management. In particular, no CMIP5 model incorporates irrigation or CA and only three CMIP6 models include irrigation, while none have CA. Pielke et al. (2011) suggested that landscape change is omitted from the CMIP5 models because the direct radiative impact of global landscape is a lower order than the radiative forcing from greenhouse gas emissions. This constitutes a reason to investigate their inclusion. That is, to distinguish between the effects of land management and other large-scale forcings such as rising $CO_2$ concentrations (Schultz et al., 2016), it is important to evaluate these processes in the GCMs and ultimately gain insight into the contrasts of impacts between regions under different climate regimes.*

*The goal of this study is thus to test the hypothesis that CESM version 1.2.2 overestimates warming trends in some regions because irrigation and CA are excluded. That is, warming rates are hypothesised to increase at a slower rate – showing signs of cooling, in irrigation- and CA-affected regions when climate models do account for a theoretical constant level of these land management practices. To realise this goal, the following objectives were formulated: (1) determine spatial warming rates using simulations that account for irrigation and CA and inspect whether CESM overestimates warming trends; (2)*

*compare the observed rates of warming to the modelled rates of warming for irrigated and CA pixels, as well as non-irrigated and non-CA pixels; and (3) estimate the impact of irrigation on the spatial average of the warming rates over time for all land, selected regions, and irrigated and CA pixels. Within this framework, the novelty of this study lies in (i) an explicit focus on land management impacts on trends as opposed to the climatology; (ii) a comparison of the subgrid versus grid-scale response, offering important new insights on the local land surface land surface response to land management; and (iii) consideration of the radiative forcing resulting from realistic land management."*

2. Unfortunately, the study does not have fine enough spatial resolution to realistically resolve these land use effects. As a result, the effects will likely be muted and quite possibly misrepresented. Even examining sub pixel (grid interval) model data is insufficient as local and mesoscale effects are missed.
As they report
*"The period 1976-2010 was simulated with a horizontal pixel resolution of 0.9° latitude × 1.25° longitude."*
This is much too coarse. Indeed since at least 4 grid increments are required to have some confidence that a feature is adequately resolved, their effective resolution is no finer than 3.6° latitude by 5° longitude.
Similarly, their observational analyses used to evaluate the model results are too coarse. They write
*"For evaluation purposes, observational datasets for annual mean T2m with a spatial resolution of 0.5◦ × 0.5◦ for the same time period were obtained from the Climate Research Unit (CRU) (Harris et al., 2014). Annual mean TXx observational datasets were obtained from the daily Global Historical Climatology Network extremes data set (GHCNDEX) (Donat et al., 2013a) and the Hadley Centre extremes data set (HadEX2) (Donat et al., 2013b) with a spatial resolution of 2.5◦ × 2.5◦ "*

**Reply.** An underlying premise of this paper is that GCMs remain the primary tool for providing long-term projected changes in the climate system and have been often used for studying land cover and land management effects on climate. Unfortunately there currently is no global model that can be run for long integrations at the spatial resolution required to fully resolve field-scale land management variations including contrasts of the irrigation/CA impact between regions under different climate regimes. So while we acknowledge that the effects play out more at the local scale and are therefore better captured with high-resolution RCMs, we believe it is still relevant to also evaluate these processes in the GCMs. To resolve this (and in combination with addressing Reviewer's point 1), we elaborate on the RCM literature and include a justification of using a GCM in the revised introduction, as follows (also detailed in paragraph 2 of our response to the Reviewer's point 1):

*"Using GCMs, such as CESM, to simulate land-atmosphere interactions for investigating the effects of irrigation and agricultural conversion has been criticized as insufficient (Niyogi et al., 2002). This is partly because their coarse resolution (e.g. of order 100 km) limits their ability to resolve land surface heterogeniety at the regional scale (Jiang et al., 2016). Furthermore, economic, societal and water resource factors are ignored – a void that initiated the so-called 'bottom-up' approach to evaluating the effects of land-use change (Douglas et al., 2006). Regarding the applicability of the knowledge produced by GCMs, they do not provide the skill required at the spatial scale to offer practical responses at the infrastructure scale (Hossain et al., 2015). Despite these constraints, GCMs remain a prime tool for projecting changes in the climate system (Fajardo et al., 2020; Gupta et al., 2020; Hofer et al., 2020). Examples include the GCMs that are part of the latest Coupled Model Intercomparison Project (CMIP6) and used by the IPCC in consecutive assessment reports (Yazdandoost et al., 2020). However, these GCMs largely exclude agricultural management. In particular, no CMIP5 model incorporates irrigation or CA and only three CMIP6 models include irrigation, while none have CA. Pielke et al. (2011) suggested that landscape change is omitted from the CMIP5 models because the direct radiative impact of global landscape is a lower order than the radiative forcing from greenhouse gas emissions. This constitutes a reason to investigate their inclusion. That is, to distinguish between the effects of land management and other large-scale forcings such as a doubling of $CO_2$ (Schultz et al., 2016), it is important to evaluate these processes in the GCMs and ultimately gain insight into the contrasts of impacts between regions under different climate regimes."*

3. And, as I mentioned above, even using sub-grid decomposition is significantly incomplete. They write
*"To examine heterogeneous influences within grid cells, subgrid tiles that represent local physical, biogeochemical, and ecological characteristics – and therefore local (subgrid) influences of irrigation and CA – were evaluated against regional (grid-scale) influences. Up to 21 surface tiles may occur within one grid cell in CLM4, including glacier, wetland, lake, urban, bare soil and 16 PFTs."*
While useful in a model sensitivity study, its lack of connection to real world data for locations where actual irrigation and conservation agriculture are occurring is a serious oversight.
In their recommendations they write
*"The findings overall emphasise the need for a more in-depth evaluation of the sensitivity of future climate projections to irrigation and CA-induced temperature changes. A sensitivity analysis, using transient irrigation and CA extents, as well as additional land management techniques and climate models based on CMIP6 output, , is recommended."*
I agree with the first sentence. The second sentence, however, is incomplete as a necessary condition. Real world testing of the skill of the models with respect to how land management affects the weather and climate is required. This must be completed using real world data that is on the appropriate space and time scales. This is not the case for this paper.

**Reply.** Thank you for raising this important point. As noted under point 1, our work goes beyond that of a model sensitivity study, as we compare our simulations against three observational products, but indeed none at the subgrid scale. We have therefore used real world data from the E-OBS European CDG dataset to conduct additional analysis. As a regional data set, it has a higher spatial resolution and therefore allows us to test the skill of the models with respect to the local effects of land management. The E-OBS data was regridded to the CESM resolution using bilinear remapping for use in this study. It captures well the MED SREX region used in this study. Please see Response Figure 2 below for the surface radiative temperature (TS) slope results over irrigated pixels (NB: this chart will not be in the revised paper but is included here to visually introduce the new observation dataset used).

[Figure]

**Response Figure 2.** *The warming trends of surface radiative temperature (TS) for the MED SREX region over irrigated pixels based on the E-OBS European CDG dataset for the period 1981-2010.*

Below are the (spatial) average of the (TS) warming rates for the MED region over (left) irrigated pixels for the irrigated and rainfed crop tiles; (right) CA pixels for the CA and CM crop tiles. Note here the slope bias has improved with the irrigated crop tile data (versus the rainfed) – see table below for data. Both of these charts have been added to Figure 8 in the paper.

[Figure]

**Response Figure 3** *(added as Figure 8e and 8f in the paper). Average of the TS warming rates over (left) irrigated pixels for the irrigated and rainfed crop tiles; (right) CA pixels for the CA and CM crop tiles. Data points specify the mean TS values within the crop tiles and pixels specified. The slope was estimated using Sen's slope for the rainfed/CM (red), irrigated/CA (blue) experiments, as well as the E-OBS European CDG dataset (green).*

The table below contains the bias and spatial RMSE of the slopes versus the E-OBS product. These results have been added to the paper as Table 2, as well as the following description and interpretation of the results. For the subgrid irrigation ($IRR_{SUB}$) ensemble, TS warming trends are overestimated by ~0.004 K yr$^{-1}$ across irrigated MED pixels, which is an improvement in terms of bias when compared to the subgrid data that does not account for irrigation (i.e. RAIN). However, according to the change in the spatial RMSE, accounting for irrigation does not improve the simulation skill for trends over MED irrigated pixels. This is likely because RMSE is more sensitive to outliers – whereas the bias is based on the spatial mean.

**Response Table 1** *(added as Table 2 in the paper). Bias and Spatial RMSE of the Ensemble Mean Warming Trends (Slopes) of the RAIN, $IRR_{SUB}$, $CA_{SUB}$ and CM Experiments Versus the E-OBS (K yr$^{-1}$) Observational Product for the years 1981-2010.*

| All MED pixels bias | | | | Irrigated MED pixels bias | | CA MED pixels bias | | All MED pixels RMSE | | | | Irrigated MED pixels RMSE | | CA MED pixels RMSE | |
|---|---|---|---|---|---|---|---|---|---|---|---|---|---|---|---|
| RAIN | CM | IRR$_{SUB}$ | CA$_{SUB}$ | RAIN | IRR$_{SUB}$ | CM | CA$_{SUB}$ | RAIN | CM | IRR$_{SUB}$ | CA$_{SUB}$ | RAIN | IRR$_{SUB}$ | CM | CA$_{SUB}$ |
| 0.032 | 0.033 | 0.022 | 0.035 | 0.015 | 0.004 | 0.013 | 0.022 | 0.040 | 0.039 | 0.031 | 0.026 | 0.028 | 0.031 | 0.027 | 0.026 |

4. They also write

*"This will support decision-making when planning land management strategies that combine resource use efficiency with climate change adaptation and mitigation, enabling sustainable intensification of land management to meet mitigation targets and future demand for food, fuel, fibre, and water."*

The authors should be made aware that there are much more inclusive tools to assess sustainability. Sensitivity results from global models is, at best, a small part on the regional and local scales. Examples of such an approach are published in

Cross, M. S., et al. (2012). "The Adaptation for Conservation Targets (ACT) framework: a tool for incorporating climate change into natural resource management." Environmental Management 50(3): 341-351. DOI: 10.1007/s00267-012-9893-7.

Hanamean, J.R. Jr., R.A. Pielke Sr., C.L. Castro, D.S. Ojima, B.C. Reed, and Z. Gao, 2003: Vegetation impacts on maximum and minimum temperatures in northeast Colorado. Meteorological Applications, 10, 203-215.

Hossain, F., E. Beighley, S. Burian, J. Chen, A. Mitra, D. Niyogi, R.A. Pielke Sr., and D. Wegner, 2017: Review approaches and recommendations for improving resilience of water management infrastructure: The case for large dams. J. Infrastructure Systems, 23, Issue 4, Dec. 2017, DOI: 10.1061/(ASCE)IS.1943-555X.0000370.

Kittel, T.G.F., et al. (2011). "A vulnerability-based strategy for incorporating climate change in regional conservation planning: Framework and case study for the British Columbia Central Interior." BC Journal of Ecosystems and Management 12(1): 7-35. http://jem.forrex.org/index.php/jem/article/view/89.

Kittel, T.G.F. 2013. "The Vulnerability of Biodiversity to Rapid Climate Change." Pp. 185-201 (Chapter 4.15), in: Vulnerability of Ecosystems to Climate, T.R. Seastedt and K. Suding (Eds.), Vol. 4 in: Climate Vulnerability: Understanding and Addressing Threats to Essential Resources, R.A. Pielke, Sr. (Editor-in- Chief). Elsevier Inc., Academic Press, Oxford. DOI: 10.1016/B978-0-12-384703-4.00437-8

Kling, M. M., Auer, S. L., Comer, P. J., Ackerly, D. D., & Hamilton, H. (2020). Multiple axes of ecological vulnerability to climate change. Global Change Biology, 26, 2798–2813

Ordonez, A., 2020. Points of view matter when assessing biodiversity vulnerability to environmental changes. *Global Change Biology*, *26*(5), pp.2734-2736.

Pielke Sr., R.A., R. Wilby, D. Niyogi, F. Hossain, K. Dairaku, J. Adegoke, G. Kallos, T. Seastedt, and K. Suding, 2012: Dealing with complexity and extreme events using a bottom-up, resource-based vulnerability perspective. Extreme Events and Natural Hazards: The Complexity Perspective Geophysical Monograph Series 196 © 2012. American Geophysical Union. All Rights Reserved. 10.1029/2011GM001086.

Romero-Lankao, P., et al. 2012: Vulnerability to temperature-related hazards: a meta-analysis and meta- knowledge approach. Glob. Environ. Change, http:// dx.doi.org/10.1016/j.gloenvcha.2012.04.002.

Stohlgren, T.J. and C.S. Jarnevich. 2009. Risk assessment of invasive species. In: M.N. Clout and P.A. Williams (eds.). Invasive Species Management: A Handbook of Principles and Techniques. New York: Oxford University Press. p. 19-35.

**Reply.** We believe that the inclusion of other tools to assess sustainability, such as the ACT framework example provided above is beyond the scope of this study and would detract from its unity. However, we do concur that the final sentence in the paper, referred to by the reviewer above, was too sweeping, and thus the final paragraph has been amended to read:

*"The findings overall provide valuable context on how model complexity can impact the simulation of trends and emphasise the need for a more in-depth evaluation of the sensitivity of future climate projections to irrigation and CA-induced temperature changes. A sensitivity analysis, using transient irrigation and CA extents, as well as additional land management techniques, within coupled climate models based on CMIP6 output, is recommended. In this way, the variance can be approximated and the relative contributions of the uncertainty sources to the total uncertainty in the model output, as well as the relative importance of irrigation and CA to the total warming trends, can be quantified and compared. If the fundamental uncertainties relating to model structure dominate, then a more detailed analysis than the regression approach used in this study is suggested. This will support decision-making on the incorporation of agricultural management processes in future GCM projects."*

5. Thus, while I am pleased to see a study examining the effects of irrigation and conservation agriculture on climate, the study has significant shortcomings as summarized in this review.

**Reply.** Thank you for your time and effort on our paper and for raising important points. We believe our resolve of these points has helped to improve our paper, which is much appreciated.

**EDITOR REPLY POINTS**

Thank you for submitting your manuscript to Earth System Dynamics. Before sending it to review, I request some initial changes to help with clarity.

1. You frame the aim of the study as being:
"to assess if climate models overestimate warming trends because theoretical constant levels of irrigation and conservation agriculture (CA) are excluded"
It does not quite make sense to me to include "theoretical constant levels of irrigation and CA" at this point - from my initial reading of the paper and understanding of CMIP5-generation models including CESM, it seems more that you are examining whether models overestimate warming trends simply because irrigation and CA are excluded, regardless of whether they are theoretically constant or not.
Also, I don't think it is appropriate to make this a statement about climate models in general, since (as you note) you are only looking at CESM.
Furthermore, since climate models tend to evolve over time and have multiple or successive versions, it is important to identify exactly which version of the model is being looked at.
So I would suggest that it would be more appropriate to say:
"to assess if CESMvn1.2, as used in CMIP5, overestimates warming trends because theoretical constant levels of irrigation and conservation agriculture (CA) are excluded"
This applies both in the abstract an in similar statements in the main text.

**Reply.** We have changed the text where necessary to specify "CESM version 1.2.2" wherever it used to say "climate models" when describing the scope of the study. We have also left out the words "theoretical constant level" in those sentences, as advised. We finally note that we run the model in AMIP mode, and therefore wish to be cautious with referencing to CMIP5.

2. Following the above, it would be useful to give more context to the extent of the discrepancy between models and observations described in the first paragraph of the introduction. Do all CMIP5 models show the discrepancy you describe? Since you are using CESM vn1.2 only, how representative is this model of family of CMIP5 models in this respect?

**Reply**: Two earlier studies sparked the idea for this research: Donat et al. (2017 GRL), showing that regional warming rates of hot extremes in CMIP5 models are inconsistent with observations across several regions, and Vogel et al. (2018 ESD), showing that some CMIP5 models exhibit unrealistically strong warming of hot extreme over Europe. Unfortunately, Donat et al. (2017) only display multi-model mean results, which do not allow us to make statements regarding the ensemble spread and the location of CESM within the ensemble. We have therefore now added more information on Vogel et al. (2018) in the introduction of the manuscript.

3. Please could you clarify in section 2.1 whether you are looking at irrigation and CA separately or together? Since your hypothesis is that their exclusion has biased the results in CMIP5 models, the reader might expect that you would look at their combined effect, but my reading of your experimental design suggests that you only applied them separately and not together - is that correct? Either way, please clarify.

**Reply**: In the first sentence of section 2.1, we have included the words "either... or" in order to clarify that irrigation and conservation agriculture were indeed modelled separately.

4. I think your figures may be be difficult to access for readers (including potential reviewers) with some types of colourblindness since red and green are often used together. I suspect that

Figures 2 and 4 may be particularly challenging. Can you use a different colour palette for Figure 2 and use symbols and/or different line styles as well as colours in Figure 4?

**Reply**: The colour palette for Figure 2 has been modified as follows.

[Figure]

*Figure 2. Added value of including irrigation and CA in the simulated warming trends over 1981-2010. Absolute change in spatial root-mean-square error (RMSE) for the (a) IRR and (b) CA ensemble relative to the CTL ensemble over different regions (x axis) and with respect to 3 observational products (y axis). Considered regions are the SREX regions where irrigation is extensive (as highlighted in the paper's Figure 1a) and where CA is extensive (see Figure 1b of the paper), in addition to global land, global irrigated land and global CA land. Observational products are for near-surface air temperature $T_{2m}$ (CRU), annual maximum daytime temperature TXx (GHCNDEX and HadEX2). The spatial RMSEs are computed for the ensemble mean warming trend in every pixel, and subsequently averaged over the selected region. Regions with an observational coverage below 50% are marked in white.*

Also, the colour palette and symbols for Figure 4 has been modified as follows:

[Figure]

[Figure]

**Figure 4.** *Spatial average of the warming rates for T$_{2m}$ (a, c and e), TXx (b, d and f), SHF (g and h), LHF (i and j) and ET (k and l) for the CESM ensembles and observations. Data points specify the mean T$_{2m}$ and TXx temperatures, SHF and LHF and ET volumes for irrigated pixels (a, b, g, I and k), CA pixels (c, d, h, j and l), and (e-f) all land pixels. The slope was estimated using Sen's slope for the CTL (red), IRR (blue), CA (cyan), CRU (purple), HadEX2 (yellow), and GHCNDEX (black) temperatures.*

5. In Figure 5, panels (a) and (b) look identical, and this is not surprising since the differences shown in panel (c) are smaller than the increments in (a) and (b). I suggest dropping panel (b).

**Reply**: Figure 5 has been updated as detailed below.

[Figure]

*Figure 5. (a) Top-of-atmosphere (TOA) net radiation $R_{n,TOA}$ [W m$^{-2}$] in the CTL ensemble. (b) Impact of irrigation on $R_{n,TOA}$. Difference (IRR-CTL) is based on the ensemble mean of each experiment for 1981–2010.*

If you are able to improve the clarity of the manuscript following the above suggestions, I will be happy to send it out for the review and discussion.

Best regards

Richard Betts (Editor)

**Reply.** Many thanks for your time and support – it is greatly valued and appreciated.

---

## Referee Report (RR1)

The authors took time to respond. Thank you.

My remaining comment is that they still make this, in my view, poor assumption:

*" An underlying premise of this paper is that GCMs remain the primary tool for providing long-term projected changes in the climate system and have been often used for studying land cover and land management effects on climate".*

Unfortunately, landscape heterogeneity, including resulting mesoscale circulation features, on spatial scales that their model cannot resolve are very important. Indeed, unless the models can resolve mesoscale features such as sea breezes, (which is an analog for landscape forced mesoscale circulations) they are going miss significant effects.  Here is one example illustrating why higher spatial resolution is essential:

Marshall, C.H. Jr., R.A. Pielke Sr., L.T. Steyaert, and D.A. Willard, 2004: The impact of anthropogenic land-cover change on the Florida peninsula sea breezes and warm season sensible weather. Mon. Wea. Rev., 132, 28-52.

A second paper, dealing with dispersion over variations in land surface forcing also shows the importance of smaller scale resolution.

Pielke, R.A. and M. Uliasz, 1993: Influence of landscape variability on atmospheric dispersion. J. Air Waste Mgt., 43, 989-994.

Even on regional scales, they will miss effects due to lack of spatial structure.

Their model can only resolve features that are at least 4 horizontal grid increments in each x-y direction. The model set up has

"The period 1976-2010 was simulated with a horizontal pixel resolution of 0.9° latitude × 1.25° longitude."

This means they can only resolve, with any fidelity, features 3.6 degrees by 5 degrees; much too coarse to resolve important landscape effects on the local, mesoscale and regional scale.

Here is an example of an analysis for aerosols and found a large potential influence on regional circulation features;

Matsui, T., and R.A. Pielke Sr., 2006: Measurement-based estimation of the spatial gradient of aerosol radiative forcing. Geophys. Res. Letts., 33, L11813, doi:10.1029/2006GL025974.

A similar large effect is expected with landscape management and landscape changes.

I recommend more text be added on this issue. One suggestion for future work is to run higher spatial resolution (both regional; mesoscale) model runs for selected shorter time slices, and compare results to the coarser resolution results they have with the global models.

After the authors and Editor consider this recommendation, I recommend acceptance.

---

## Author Response (AR2)

**Revision comment on* "Agricultural management effects on mean and extreme temperature trends" *by* Aine M. Gormley-Gallagher et al.**

**Anonymous Referee #2**

Received: 30 October 2021

The authors took time to respond. Thank you.

**Reply.** The Reviewer's time and recent recommendation are greatly appreciated. We have addressed the reviewer's recommendation as detailed below (in blue) and the amended manuscript text is provided in italics.

1. My remaining comment is that they still make this, in my view, poor assumption:

   > *"An underlying premise of this paper is that GCMs remain the primary tool for providing long-term projected changes in the climate system and have been often used for studying land cover and land management effects on climate".*

   Unfortunately, landscape heterogeneity, including resulting mesoscale circulation features, on spatial scales that their model cannot resolve are very important. Indeed, unless the models can resolve mesoscale features such as sea breezes, (which is an analog for landscape forced mesoscale circulations) they are going miss significant effects. Here is one example illustrating why higher spatial resolution is essential:

   - Marshall, C.H. Jr., R.A. Pielke Sr., L.T. Steyaert, and D.A. Willard, 2004: The impact of anthropogenic land-cover change on the Florida peninsula sea breezes and warm season sensible weather. Mon. Wea. Rev., 132, 28-52.

   A second paper, dealing with dispersion over variations in land surface forcing also shows the importance of smaller scale resolution.

   - Pielke, R.A. and M. Uliasz, 1993: Influence of landscape variability on atmospheric dispersion. J. Air Waste Mgt., 43, 989-994.

   Even on regional scales, they will miss effects due to lack of spatial structure. Their model can only resolve features that are at least 4 horizontal grid increments in each x-y direction. The model set up has:

   > *"The period 1976-2010 was simulated with a horizontal pixel resolution of 0.9° latitude × 1.25° longitude."*

   This means they can only resolve, with any fidelity, features 3.6 degrees by 5 degrees; much too coarse to resolve important landscape effects on the local, mesoscale and regional scale. Here is an example of an analysis for aerosols and found a large potential influence on regional circulation features;

   - Matsui, T., and R.A. Pielke Sr., 2006: Measurement-based estimation of the spatial gradient of aerosol radiative forcing. Geophys. Res. Letts., 33, L11813, doi:10.1029/2006GL025974.

   A similar large effect is expected with landscape management and landscape changes.

   I recommend more text be added on this issue. One suggestion for future work is to run higher spatial resolution (both regional; mesoscale) model runs for selected shorter time slices, and compare results to the coarser resolution results they have with the global models.

   After the authors and Editor consider this recommendation, I recommend acceptance.

**Reply.** To the introductory paragraph that focuses on the limitations of GCMs (paragraph 4), we have added:

> *"Regarding the applicability of the knowledge produced by GCMs, they do not provide the skill required at the spatial scale to offer practical responses at the infrastructure scale (Hossain et al., 2015) or in terms of water resource management (Marshall et al., 2004)".*

The following reference has thus been added to the reference list:

> *"Marshall, C.H. Jr., Pielke Sr., R. A., Steyaert, L. T. and Willard, D. A. 2004: The impact of anthropogenic land-cover change on the Florida peninsula sea breezes and warm season sensible weather. Monthly Weather Review, 132, 28-52."*

Also, to the conclusion, the following text has been added:

> *"Furthermore, we encourage the community to compare the coarser resolution results gained in this GCM study with higher spatial resolution models and for seasonal and monthly time periods. This will support decision-making on the incorporation of agricultural management processes in future GCM projects."*

Lastly, the final sentence in the abstract has been adjusted to now read:

> *"Our results underline that agricultural management has complex and nonnegligible impacts on the local climate and highlight the need to evaluate the representation of land management in global climate models using climate models of higher resolution."*

We thank you for your continued effort on our paper and for ensuring that important information is included. We believe your contribution and our response has greatly helped to improve the value and quality of our paper.